# eIF4E-independent translation is largely eIF3d-dependent

Mykola Roiuk [1,2,3], Marilena Neff[1,2,3] & Aurelio A. Teleman [1,2,3] ✉

Translation initiation is a highly regulated step needed for protein synthesis. Most cell-based mechanistic work on translation initiation has been done using non-stressed cells growing in medium with sufficient nutrients and oxygen. This has yielded our current understanding of 'canonical' translation initiation, involving recognition of the mRNA cap by eIF4E1 followed by successive recruitment of initiation factors and the ribosome. Many cells, however, such as tumor cells, are exposed to stresses such as hypoxia, low nutrients or proteotoxic stress. This leads to inactivation of mTORC1 and thereby inactivation of eIF4E1. Hence the question arises how cells translate mRNAs under such stress conditions. We study here how mRNAs are translated in an eIF4E1-independent manner by blocking eIF4E1 using a constitutively active version of eIF4E-binding protein (4E-BP). Via ribosome profiling we identify a subset of mRNAs that are still efficiently translated when eIF4E1 is inactive. We find that these mRNAs preferentially release eIF4E1 when eIF4E1 is inactive and bind instead to eIF3d via its cap-binding pocket. eIF3d then enables these mRNAs to be efficiently translated due to its cap-binding activity. In sum, our work identifies eIF3d-dependent translation as a major mechanism enabling mRNA translation in an eIF4E-independent manner.

Translation initiation is a regulated and rate-limiting step in mRNA translation, thereby controlling the synthesis of proteins. Historically, most cell-based mechanistic work on translation initiation has been done using non-stressed cells growing in a medium with sufficient nutrients and oxygen. This has yielded our current understanding of 'canonical' translation initiation, which starts with recognition of the 5' cap on an mRNA by eIF4E1. eIF4E1 then binds and recruits eIF4G1, which in turn leads to the recruitment of multiple initiation factors and the small subunit of the ribosome[1–5]. Many cells, however, are exposed to stress. For instance, cells inside poorly vascularized regions of a tumor are hypoxic and have low nutrients, leading to inactivation of mTORC1 and to activation of the Integrated Stress Response[6–10]. When mTORC1 is inhibited, this results in activation of eIF4E-binding protein (4E-BP) and consequently sequestration and inhibition of eIF4E1[11]. Hence, this raises the question of how stressed cells translate mRNAs using non-canonical mechanisms that do not depend on eIF4E1.

Several eIF4E1-independent initiation mechanisms have been described that act on specific mRNAs or under certain conditions. For instance, upon hypoxia, a complex consisting of HIF-2α, RBM4, and eIF4E2 binds the cap on specific mRNAs and activates their translation[12,13]. NCBP1/3 was shown to be crucial for stress-resistant translation of JUND[14]. Another mechanism, which is especially important in response to heat shock, involves N6-methyladenosine (m6A) in the 5'UTR of certain mRNAs which guides 5'-end-dependent but cap-independent translation initiation[15,16]. Translation from Internal Ribosome Entry Sites (IRESs) is also known to be cap-independent[17]. Finally, a component of the eIF3 complex, eIF3d, has been shown to have cap-binding capacity and to associate with DAP5/eIF4G2/NAT1 to promote translation of c-Jun, mRNAs encoding transcription factors and regulators or the epithelial-to-mesenchymal transition (EMT)[18–22], and genes needed for differentiation of Treg cells[23]. Although each of these mechanisms can act independently of eIF4E, a global analysis of eIF4E-

[1]German Cancer Research Center (DKFZ) Heidelberg, Heidelberg, Germany. [2] Faculty of Medicine, Heidelberg University, Heidelberg, Germany. [3]Faculty of Biosciences, Heidelberg University, Heidelberg, Germany. ✉e-mail: a.teleman@dkfz.de

independent translation is lacking, as is an understanding of whether one of these pathways is predominant when eIF4E1 levels are limiting.

We aimed here to understand how cells translate mRNAs when eIF4E1 is inhibited. To this end, we devised a setup to specifically block eIF4E1 function by expression a version of eIF4E-binding protein (4E-BP) that is constitutively active. Ribosome profiling in this condition identified a number of mRNAs that are still well translated when eIF4E1 is inactive. We show that these mRNAs are translated in a 5′-end-dependent and cap-dependent but eIF4E-independent manner. When eIF4E is inactive, we find that these mRNAs preferentially release eIF4E1 and bind instead to eIF3d via its cap-binding pocket. Indeed, the ability of these mRNAs to escape the general downregulation of translation upon eIF4E1 inactivation depends on eIF3d and its ability to bind cap. In sum, our work identifies eIF3d-dependent translation as a major mechanism enabling persistent mRNA translation in an eIF4E-independent manner.

## Results

### Global translation drops to roughly 30% upon eIF4E inhibition

Consistent with previous studies, strong inhibition of mTORC1 and mTORC2 with Torin leads to a 60% drop in global mRNA translation, quantified via incorporation of O-propargyl-puromycin (OPP) (assuming cycloheximide treatment, CHX, completely blocks translation and thus represents background signal, Suppl. Fig. 1A). Of note, the drop in global translation caused by Torin is not as strong as that caused by cycloheximide, indicating that some translation still occurs when mTORC1 is inactive. One downstream consequence of mTORC1 inhibition is the dephosphorylation and activation of eIF4E-binding proteins (4E-BPs), which bind and sequester eIF4E (Suppl. Fig. 1B). Hence, to specifically study this aspect of the stress response, we expressed in HeLa cells a constitutively-active version of 4E-BP where the 4 main mTORC1 phosphorylation sites (Thr37/46, Ser65, Thr70) were mutated to alanine (4E-BP-4A). Cap pull-downs confirmed that expression of 4E-BP-4A strongly reduces the amount of eIF4G binding to eIF4E (Suppl. Fig. 1C, D), to a degree similar to that caused by Torin (Suppl. Fig. 1C, D). We did not observe any increase in eIF2alpha phosphorylation (Suppl. Fig. 1E, F) suggesting that 4E-BP-4A overexpression does not activate the integrated stress response. In contrast, as expected, 4E-BP-4A overexpression reduces eIF4E phosphorylation my MNK (Suppl. Fig. 1E), since MNK is recruited to eIF4E via eIF4G binding[24]. 4E-BP-4A overexpression causes a 70% drop in global translation levels (assuming CHX treatment = no translation, Suppl. Fig. 1G, H), and a 50% drop in polysome amounts (Suppl. Fig. 1I, J), similar in magnitude to Torin treatment (Suppl. Fig. 1A). In sum, 4E-BP-4A overexpression phenocopies the eIF4E inhibition observed upon strong mTORC1 inhibition.

### Specific mRNAs escape the translational repression caused by eIF4E inhibition

To study the impact of eIF4E inhibition on translation of individual mRNAs, we performed ribosome profiling[25] on HeLa cells over-expressing 4E-BP-4A versus control HeLa cells transfected with empty plasmid (Fig. 1A). Amongst the mRNAs that have reduced translation upon 4E-BP-4A expression are those encoding ribosomal proteins and translation initiation factors (not shown). In contrast, translation of specific mRNAs such as CDKN2B and RPP25 are 4–5 fold higher than the average mRNA in the cell (red dots, Fig. 1A). Likewise, translation of CDKN1B was higher, consistent with a previous study[26]. Note that ribosome profiling normalizes away the global drop in translation caused by 4E-BP-4A expression (Suppl. Fig. 1G–J). Hence, an increase in translation efficiency actually means that CDKN2B and RPP25 do not follow the general drop in translation, resulting in translation levels that are similar to non-stressed conditions or mildly increased. We, therefore, refer to these mRNAs indicated in red in Fig. 1A as "resistant" mRNAs. We validated the ribosome profiling results in two ways. First,

we detected the localization of endogenous mRNAs in polysome gradients following 4E-BP1-4A expression. 4E-BP1-4A causes a strong reduction in polysomes (Fig. 1B) and consistent with this, a shift away from polysomes and into monosomes of mRNAs which did not show an increase in translation efficiency in the ribosome footprinting (Fig. 1A) such as GLO1, RPL13A, or GAPDH (Fig. 1C). In contrast, 'resistant' mRNAs such as CDKN2B or RPP25, DYNC1H1 either stayed in the heavy polysome fractions or even shifted into heavier fractions compared to the control condition (Fig. 1C) indicating continued association with ribosomes. Secondly, we tested protein synthesis upon 4E-BP-4A expression. To specifically assay the rate of de novo protein synthesis, we used the BONCAT approach, involving a pulse of metabolic labeling with 4-azido-L-homoalanine (AHA), which incorporates into nascent polypeptides instead of methionine[27] (Suppl. Fig. 2A). AHA can then be pulled-down by click chemistry to assay the proteins synthesized during the AHA pulse. Since this method requires the removal of methionine from the medium for 30 min prior to the addition of AHA, we first confirmed that methionine removal does not cause a drop in mTORC1 activity within this timeframe (Suppl. Fig. 2B, C), in agreement with mTORC1 acutely sensing leucine, glutamine, and arginine[28]. Methionine removal for 60 min also does not change global levels of ubiquitination (Suppl. Fig. 2D, E). In agreement with OPP incorporation, total AHA incorporation also revealed a drop in translation caused by 4E-BP-4A expression (Suppl. Fig. 2F, G). We then assayed the de novo synthesis of specific proteins by immuno-blotting the AHA pulldown. Although 4E-BP-4A overexpression caused a drop in translation of all the proteins we assayed (Fig. 1D, E), translation of the 'resistant' mRNAs (RPP25, COL12A1, TXNIP, etc) dropped less than other mRNAs such as GAPDH, RPS6, or RPS15 (Fig. 1D, E). Thus, although the absolute effect sizes detected by polysome profiling (Fig. 1B, C) versus BONCAT (Fig. 1D, E) do not match perfectly (the 'resistant' mRNAs stay in heavy polysomes or even shift into heavier polysomes upon 4E-BP-4A expression, whereas their translation levels by BONCAT drop), nonetheless, the relative effect of 4E-BP-4A on translation of 'resistant' mRNAs versus other mRNAs is consistent across all three methods - ribosome profiling, polysome profiling, and BONCAT.

### Resistant genes continue being translated upon glucose starvation

One stress that inhibits mTORC1 is glucose starvation (Suppl. Fig. 3A)[29]. We, therefore, asked if the mRNAs we identified as being resistant to eIF4E inhibition will continue being translated when cells are subjected to glucose starvation. As expected, glucose starvation causes a mild but incomplete reduction in polysomes, since cells in culture can also metabolize glutamine (Suppl. Fig. 3B). We then quantified the localization of endogenous mRNAs in the polysome gradient by Q-RT-PCR. This revealed that control mRNAs such as GAPDH, RPL13A, or GLO1 shift away from polysomes and into monosomes upon glucose starvation, indicative of reduced translation (Suppl. Fig. 3C). In contrast, 'resistant' mRNAs such as CDKN2B, RPP25, or DYNC1H1 either do not shift upon glucose starvation, or shift mildly into heavier polysome fractions, indicating that their engagement with ribosomes does not decrease (Suppl. Fig. 3C). In sum, the mRNAs we identified as being resistant to eIF4E inhibition are also resistant to a physiologically relevant stress, glucose starvation.

### Resistant genes are enriched for growth and cell cycle inhibitors

We next used Gene Set Enrichment Analysis to test whether mRNAs involved in particular molecular or cellular functions are enriched amongst the mRNAs whose translation is either sensitive or resistant to eIF4E inhibition (Suppl. Fig. 4). mRNAs encoding factors involved in translation and growth are down-regulated (Suppl. Fig. 4B). In contrast, negative regulators of growth and the cell cycle are enriched amongst the up-regulated mRNAs (Suppl. Fig. 4A), such as CDKN1B.

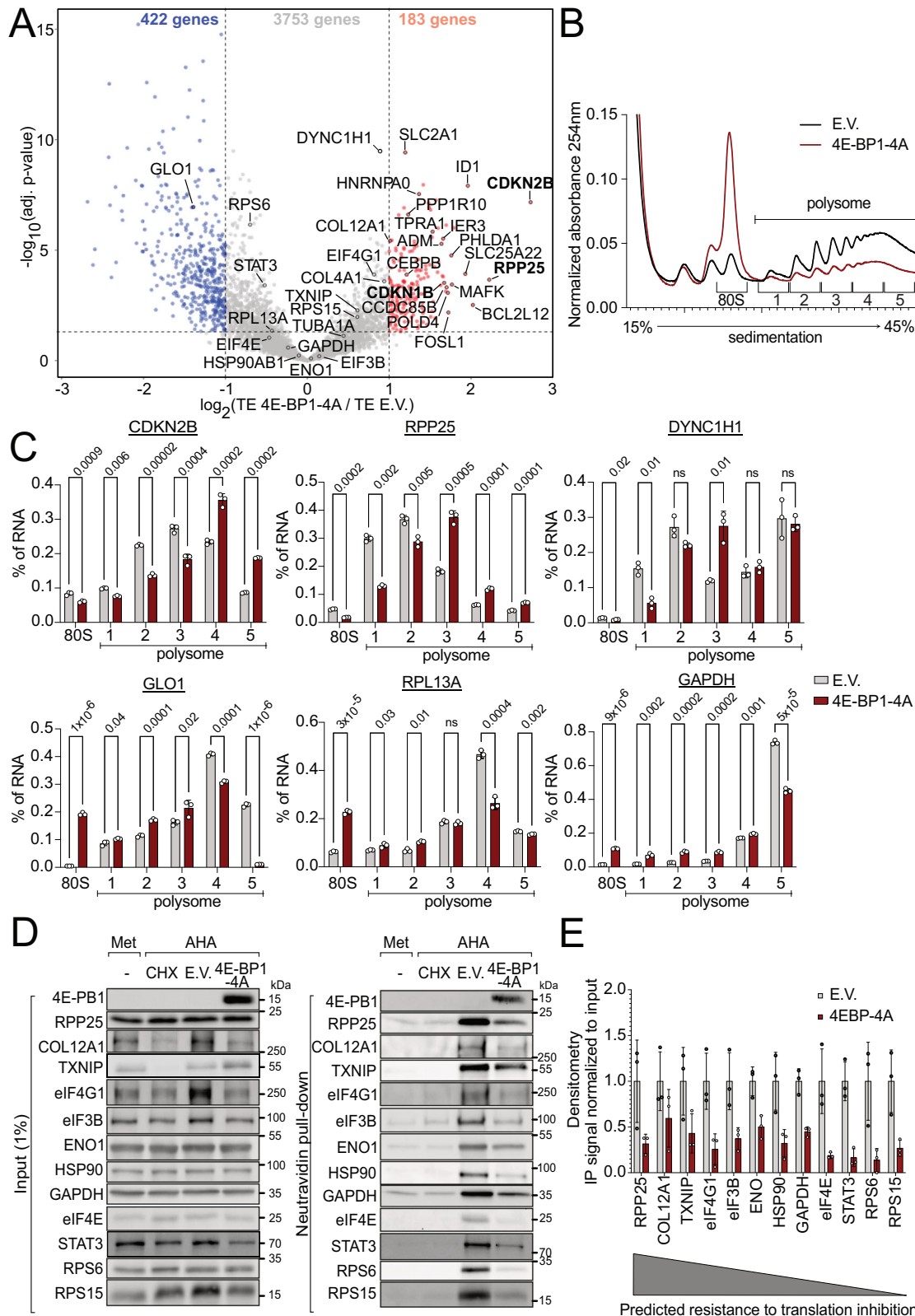

This is expected because the cell cycle needs to be blocked when mTORC1 activity is reduced. Translation of organic transporters was also increased, perhaps because low mTORC1 signals to cells a lack of nutrients so cells might respond by trying to increase nutrient uptake. Finally, translation of extracellular matrix components and adhesion molecules were also increased, although the biological significance of this is unclear.

## TOS-less 4E-BP blocks eIF4E without inhibiting mTORC1

We were surprised to find that many of the mRNAs with reduced translation upon 4E-BP-4A expression were 5′TOP mRNAs such as mRNAs encoding for ribosomal proteins (Suppl. Fig. 4B) since 5′TOP mRNAs are thought to be suppressed by Larp1 when mTORC1 activity is reduced[30,31]. Hence, we tested if 4E-BP-4A expression reduces mTORC1 activity, and found that this is indeed the case, seen as a

**Fig. 1 | Ribosome profiling identifies mRNAs resistant to eIF4E inhibition.**
**A** Scatter plot of log2 (fold change of Translation Efficiency 4E-BP1-4A/E.V.) versus
significance. Significant candidates with log2(fold change) > 1 are red, <−1 are blue.
Candidates used for reporters and for validation of the dataset are labeled with
gene names. Significance was estimated with the Wald test performed by DESeq2
package, *p*-values adjusted for multiple comparison. **B, C** Validation of the ribo-
some profiling from **A** by qRT-PCR for endogenous mRNAs in polysome gradients.
**B** Representative example of polysome gradients from HeLa overexpressing either
4E-BP1-4A or empty vector (E.V.). The "80S" and "polysome" fractions were col-
lected as indicated. An equal amount of RLuc mRNA was spiked into each fraction
to enable comparison of transcript distribution between fractions. **C** RNA quanti-
fication by Q-RT-PCR from 3 biological replicates. Distribution of resistant *CDKN2B,
RPP25, DYNHC1H1* and sensitive *GLO1, RPL13A, GAPDH* transcripts is depicted.
**D, E** Validation of the ribosome profiling by BONCAT. **D** One representative replica
of a BONCAT de novo protein synthesis assay. **E** Quantification of three indepen-
dent biological replicates. The transcripts are sorted in order of decreasing resis-
tance to 4E-BP-induced translation inhibition as predicted by the footprinting. All
panels: data are presented as mean values, error bars=std. dev., Significance by
unpaired, two-sided, *t*-test adjusted for multiple testing. ns=not significant.

decrease in phosphorylation of the direct target pS6K (Suppl. Fig. 5A).
We hypothesized 4E-BP-4A might inhibit mTORC1 by binding to it and
titrating it away from other substrates, similar to how PRAS40 over-
expression acts[32]. We, therefore, mutated *4E-BP-4A* to delete the TOS
motif which mediates binding to the mTORC1 complex[33], yielding 4E-
BP-4AΔTOS. We found that 4E-BP-4AΔTOS was a more specific inhi-
bitor of eIF4E1 than 4E-BP-4A, since it does not inhibit mTORC1 (Suppl.
Fig. 5A), yet it still blocks eIF4G binding to eIF4E (Suppl. Fig. 5B) and
reduces mRNA translation (Suppl. Fig. 5C, D). Thus, we used 4E-BP-
4AΔTOS for all further experiments in this manuscript, unless other-
wise stated.

### Resistance to eIF4E inhibition is often a property of the 5′UTR
We next looked for features that might cause the translation of certain
mRNAs to be resistant to eIF4E inhibition. This revealed that 'resistant'
mRNAs tend to have longer 5′UTRs (Suppl. Fig. 6A). In contrast, there
was no clear correlation between CDS length or 3′UTR length and
resistance to eIF4E inhibition (Suppl. Fig. 6B, C). To test whether ele-
ments in the 5′UTRs of the 'resistant' mRNAs enable them to continue
being translated when eIF4E is inhibited, we cloned the 5′UTRs of 18 of
the most 'resistant' mRNAs into Renilla luciferase (RLuc) reporter
plasmids. For all luciferase assays, we co-transfected the RLuc repor-
ters with a Firefly luciferase (FLuc) normalization control. As a negative
control we used the 5′UTR of the hemoglobin B (*HBB*) gene, which is
known to be translated exclusively via the 5′-cap[34] and highly eIF4E
dependent[35]. This revealed that many of the tested 5′UTRs caused
translation of RLuc to become resistant to 4E-BP-4A overexpression
(Fig. 2A), to *eIF4E* knockdown (Suppl. Fig. 7A-D), or to Torin treatment
(Suppl. Fig. 7E). Note that in these experiments we calculate the ratio of
RLuc reporter luminescence counts to FLuc normalization control
counts. Hence, this is a relative measure - if translation of the RLuc
reporter drops less than translation of the FLuc normalization control
reporter upon 4E-BP-4A expression, this yields a value greater than 1.
For the reporters that did not show resistance, it is possible that the
gene has different splice isoforms and we did not clone the relevant 5′
UTR. To confirm that these effects are translational, we quantified
RLuc counts normalized to reporter mRNA levels by Q-RT-PCR and
found that indeed 4E-BP-4A expression reduced translation of the *HBB*
control reporter whereas the *CDKN2B, RPP25, BCL2L12*, and *SLC2A1*
reporters were more resistant (Suppl. Fig. 7F). We observed a similar
pattern of reporter resistance to 4E-BP expression in HEK293T cells
and in U2OS cells, indicating this is a general phenomenon in human
cells (Suppl. Fig. 7G-H). Inhibition of MNK-dependent phosphorylation
of eIF4E did not phenocopy 4E-BP-4A overexpression or *eIF4E* knock-
down (Suppl. Fig. 8) indicating that eIF4E phosphorylation is not
relevant for translation of the resistant mRNAs. In sum, we conclude
that in many cases, 'resistant' mRNAs contain sequence elements in
their 5′UTRs enabling them to continue being translated when eIF4E is
inactivated. We thus selected for further study the RLuc reporters
which are resistant to eIF4E inhibition.

### Resistance to eIF4E inhibition does not require IRES activity
We next sought to understand the molecular mechanisms enabling
these mRNAs to be resistant to eIF4E inhibition. One translation

initiation mechanism that is cap-independent is translation via an
Internal Ribosome Entry Site (IRES)[17,36]. Although it is controversial
whether IRESs exist on cellular mRNAs[37], we tested whether the
eIF4E-independent 5′UTRs display IRES activity. As described below,
however, we did not obtain evidence indicating that these resistant
mRNAs are translated in an IRES-dependent manner. We cloned
these 5′UTRs into a bicistronic reporter, placing them between the
upstream RLuc and the downstream FLuc coding sequences
(Fig. 2B), in vitro transcribed the reporters to avoid well-known
artefacts that can occur if such reporters are transfected as plasmid
DNA[38], and transfected them into cells (Fig. 2B). As a positive control
for IRES activity, we used the encephalomyocarditis virus (EMCV)
IRES. This revealed that almost all the tested 5′UTRs had no sig-
nificant IRES activity compared to the *HBB* 5′UTR (Fig. 2B), except
the 5′UTR of *CDKN1B*. We also tested whether a putative IRES activity
of these 5′UTRs responds to 4E-BP activation (Suppl. Fig. 9A). As
previously reported[39,40], translation from the EMCV IRES increases
mildly upon eIF4E inhibition, perhaps due to translation initiation
factors and ribosomal subunits becoming liberated when cap-
dependent translation is shut off (Suppl. Fig. 9A). In contrast,
translation of none of the reporters containing 5′UTRs from resis-
tant genes increased upon 4E-BP-4A expression, or behaved differ-
ently from the *HBB* negative control (Suppl. Fig. 9A). In conclusion,
the 5′UTRs of these resistant genes do not appear to be resistant due
to IRES activity. Consistent with this, we tested the effect of intro-
ducing a stem-loop at the 5′-end of the reporters, near the cap
(Fig. 2C). Such a stem-loop has previously been shown to reduce cap-
dependent translation but not IRES translation, which involves
internal entry[41]. As expected, introduction of the 5′ stem-loop
decreased expression of the cap-dependent *HBB* negative control
reporter, but not a reporter containing the EMCV IRES (Fig. 2C). The
vast majority of the reporters carrying 5′UTRs of the 'resistant' genes
behaved like the *HBB* negative control, dropping in expression upon
introduction of the stem-loop (Fig. 2C). For the reporters that were
not so efficiently suppressed by the stem-loop in this plasmid-based
setup, we repeated the assay with mRNA transfections and observed
a strong drop with all reporters apart from the positive control
reporter carrying the EMCV IRES, which increased upon Torin co-
treatment (Suppl. Fig. 9B). Thus, most of the translation on these
"resistant" 5′UTRs appears to be IRES-independent and 5′-end
dependent.

### Resistance to eIF4E inhibition does not require m6A
Another translational mechanism that is eIF4E-independent is a
newly described mechanism that is m6A-dependent and cap-
independent[15,42,43]. Since mTORC1 modulates m6A levels[44,45], we
tested whether this mechanism is responsible for the translation of
the 'resistant' mRNAs. To this end, we inhibited m6A methylation
either pharmacologically with STM2457 for 16 h, or by knocking
down the m6A writer *METTL3* (Suppl. Fig. 9C–F). Neither treatment,
however, reduced the resistance of the luciferase reporters to 4E-BP-
4A expression (Suppl. Fig. 9C–F) indicating that the mechanism
responsible for their translation when eIF4E is inhibited is m6A-
independent.

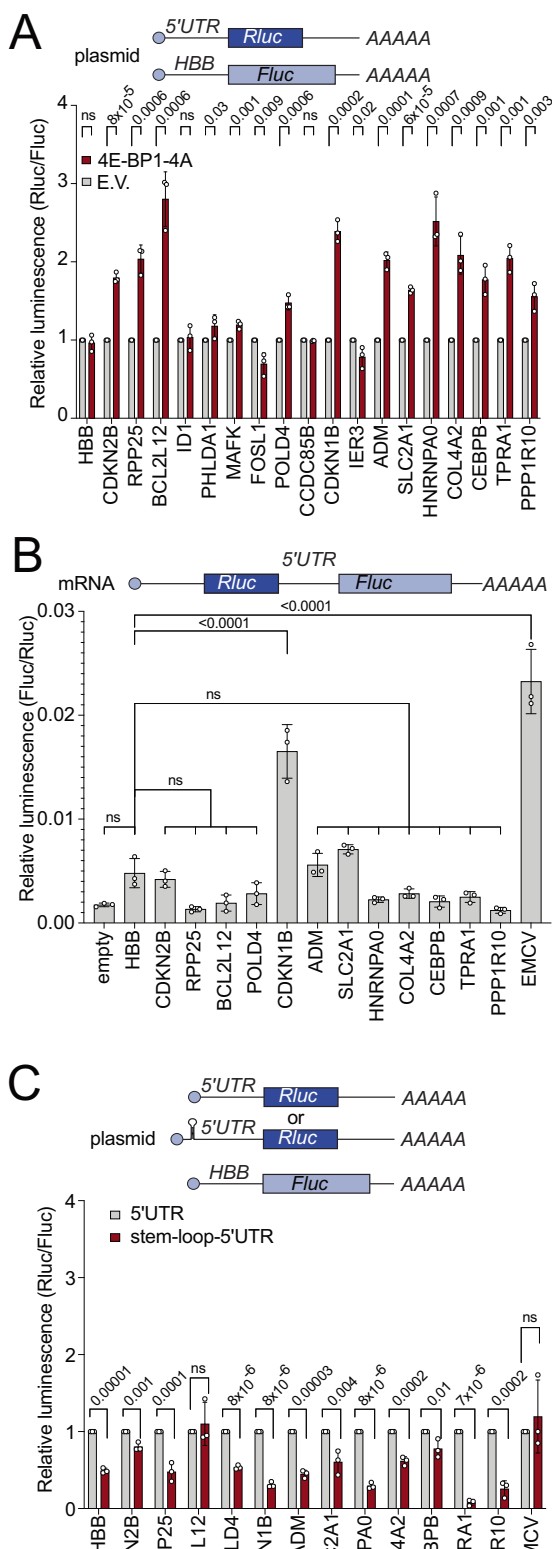

**Fig. 2 | Translation of resistant transcripts is 5′-end dependent and IRES-independent. A** The 5′UTRs of mRNAs resistant to eIF4E inhibition are often sufficient to impart resistance to a luciferase reporter. Reporters carrying the 5′UTRs of the indicated candidate genes were cloned upstream of Renilla Luciferase (RLuc) and co-transfected with a Firefly Luciferase (FLuc) normalization control. Both the negative control RLuc reporter and the FLuc normalization control carry the 5′UTR of beta globin (HBB). eIF4E was inhibited by co-transfecting with 4E-BP1-4A. (E.V.=empty vector). **B** An mRNA-based bicistronic assay reveals little or no IRES activity in the 5′UTRs of resistant mRNAs. Only the positive control EMCV IRES and the 5′UTR of *CDKN1B* showed IRES activity significantly above background. **C** Most reporters carrying the 5′UTRs of resistant mRNAs are translated in a 5′-end dependent manner, assayed by introducing a stable stem-loop at their 5′-end. The stable stem-loops blunts translation for the vast majority of reporters. The EMCV IRES serves as a positive control for a 5′-end independent reporter. All panels: n = 3 biological replicates. Data are presented as mean values, error bars=std. dev., significance by unpaired, two-sided, *t*-test adjusted for multiple testing (**A**, **C**) or by Tukey's multiple comparison test ANOVA (**B**). ns not significant.

cell[48]. We then transfected these mRNAs into cells and found that the vast majority of the translation of these mRNAs was cap-dependent (Fig. 3A). Hence the small amount of translation that is cap-independent cannot account for their translation when eIF4E is inhibited. Indeed, consistent with this, the A-capped reporters were no longer resistant to Torin-mediated eIF4E inhibition, instead behaving similar to the negative control *HBB* reporter (Fig. 3A, B).

As an additional test of cap dependence, we generated a system where the uncapped reporter would be produced inside the cell. We cloned the *RPP25* reporter downstream of the U6 promoter, which recruits Polymerase III and results in an uncapped transcript. To terminate transcription, Pol III needs to encounter a stretch of 4 or more poly-T, therefore we introduced synonymous mutations in the Rluc ORF to remove poly-T stretches and introduced a long poly-T stretch after the region encoding the 3′UTR (Fig. 3C). Finally, to enable the export of the uncapped construct from the nucleolus to the cytosol, we introduced in the 3′-UTR a viral constitutive transport element (CTE-element) that permits TAP-mediated nuclear export of uncapped RNA[49,50]. As expected, this construct displays low activity which is nonetheless higher than the activity of untransfected cells (Fig. 3C). Unlike the capped *RPP25* reporter, the uncapped *RPP25* reporter is not resistant to 4E-BP-4A expression (Fig. 3D), confirming the necessity of m7G-cap for the eIF4E-resistant mechanism.

### Resistant mRNAs let go of eIF4E upon eIF4E inhibition

We hypothesized two opposing scenarios for how some mRNAs might be resistant to eIF4E inhibition. One scenario is that upon eIF4E inhibition, resistant mRNAs compete better for the little remaining eIF4E activity in a cell compared to all other mRNAs - e.g. they bind eIF4E more strongly and/or are able to retain some eIF4E-dependent translation initiation. The opposite scenario is that resistant mRNAs can also be translated in an eIF4E-independent manner due to another cap-binding translation initiation factor, in which case they should release eIF4E upon eIF4E inhibition. To distinguish these two possibilities, we performed an eIF4E pulldown in the presence or absence of 4E-BP-4A (Fig. 4A and Suppl. Fig. 10) and sequenced the associated mRNAs (Fig. 4B). (This was done using full-length 4E-BP1-4A in order to correlate these results to the riboseq data presented in Fig. 1). This revealed that the 'resistant' mRNAs (shown in yellow) such as *RPP25* or *CDKN2B* were amongst the mRNAs that showed the largest drop in eIF4E binding upon 4E-BP-4A expression (Fig. 4B, C). We confirmed this by Q-RT-PCR on eIF4E IPs from cells expressing 4E-BP-4A or treated with Torin (Fig. 4D). Indeed, transcriptome-wide, we observed a negative correlation between the change in translation efficiency of mRNAs upon eIF4E inhibition and their binding to eIF4E (Fig. 4E). In sum, this suggests that 'resistant' mRNAs have an increased capacity to shift from binding eIF4E to binding another cap-binding protein when eIF4E is inhibited.

### eIF4E-independent translation is cap-dependent

A number of different proteins have been reported to bind the mRNA cap in addition to eIF4E[46,47], hence mRNA translation can be eIF4E-independent but still cap-dependent. To test this, we in vitro transcribed the 'resistant' luciferase reporters with either a standard 7-methylguanylate cap, or with an unfunctional analogous ApppG 'A-cap' that is not recognized by eIF4E yet provides stability to mRNAs in a

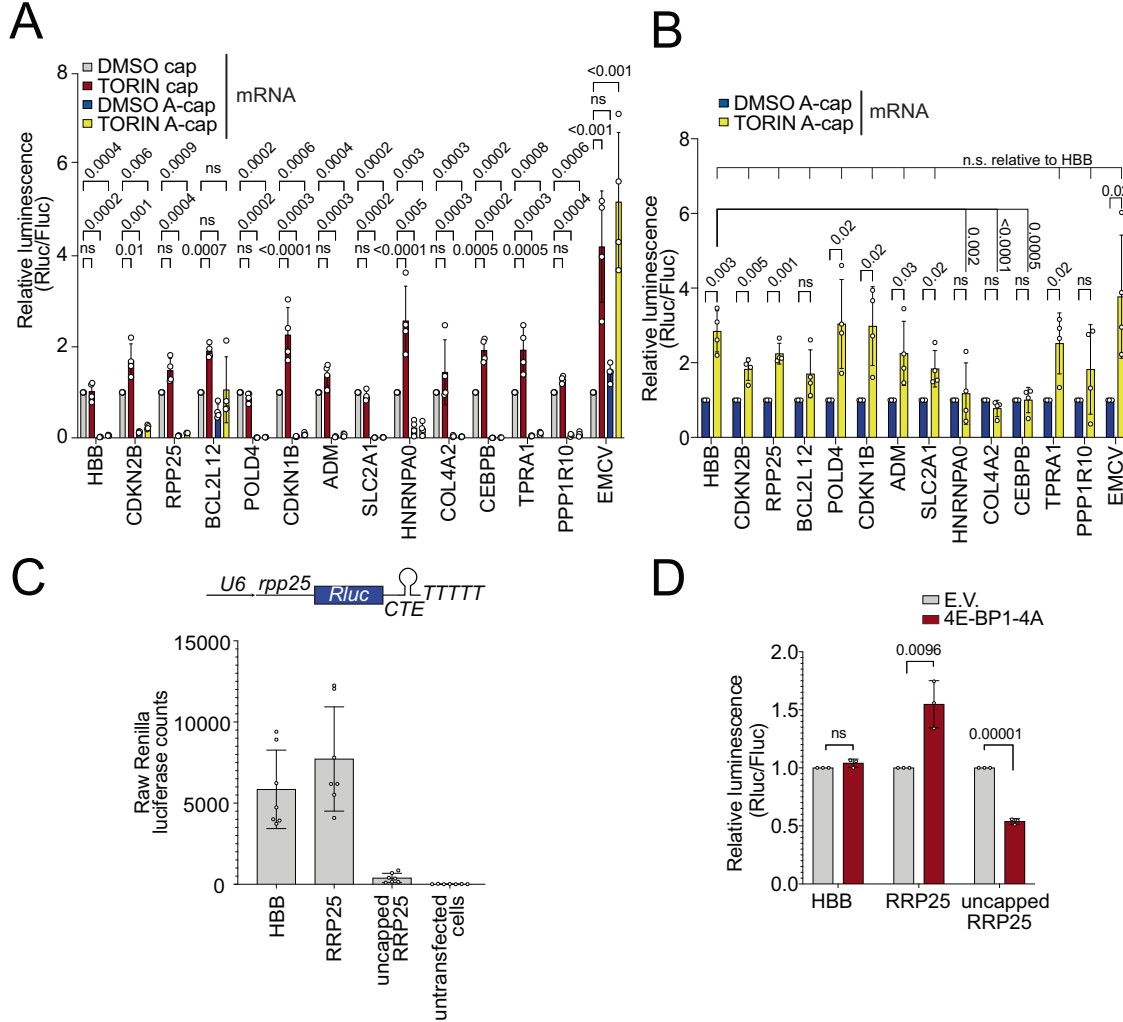

**Fig. 3 | Translation of resistant transcripts is cap-dependent. A, B** The vast majority of translation of the resistant reporters requires cap. (A) Capped and A-capped mRNA reporters were transfected into HeLa cells and co-treated either with DMSO or TORIN for 6 h. EMCV IRES serves as positive control for cap-independent translation. **B** Only the only A-capped reporters from **A** are shown, with the DMSO condition normalized to 1. **C** Translation of the *RPP25* reporter requires cap. Cells were transfected either with the standard *RPP25* reporter, or with a *RPP25* reporter driven by the U6 promoter which does not generate capped transcripts. The raw luciferase counts are shown. **D** Resistance of the *RPP25*

reporter to eIF4E inhibition requires cap. Cells were transfected either with standard *HBB* or *RPP25* reporters which generated capped transcripts in cells, or with the uncapped *RPP25* reporter illustrated in **C**, together with either 4E-BP1-4A or empty vector (E.V.) as a control. All panels: data are presented as mean values, error bars=std. dev.. panel A and B $n = 4$ biological replicates, panel **C** $n = 7$ technical replicates, panel **D** $n = 3$ biological replicates. Significance by unpaired, two-sided, *t*-test adjusted for multiple testing (**D**) or by Dunnett's multiple comparison test ANOVA (**A**, **B**). ns not significant.

## Resistance to eIF4E inhibition does not depend on eIF4E2, eIF4E3, eIF3l, or a battery of proteins reported to bind the cap

We next aimed to identify the alternate cap-binding protein that enables 'resistant' mRNAs to be translated when eIF4E is inhibited. There are three members of the eIF4E family - eIF4E1, eIF4E2, and eIF4E3. Compared to the canonical cap-binding member eIF4E1, the other two have significantly lower cap-binding affinity but nonetheless participate in regulating translation initiation in specialized circumstances[13,51–53]. To test the role of eIF4E2, we generated *eIF4E2*-knockout HeLa cells (Suppl. Fig. 11A-A') but the 'resistant' luciferase reporters still showed a relative increase in expression upon 4E-BP-4A expression also in *eIF4E2*-KO cells (Suppl. Fig. 11B) indicating that eIF4E2 is dispensable. For eIF4E3, we could not detect any protein in HeLa cells by immunoblotting (Suppl. Fig. 11C) in agreement with previously published mass spectrometry results (Suppl. Fig. 11D)[54]. Nonetheless, we could detect some endogenous *eIF4E3* mRNA by Q-RT-PCR which decreased substantially if cells were transfected with an sgRNA targeting *eIF4E3* (Suppl. Fig. 11E). However, this sgRNA did not

affect the ability of the resistant luciferase reporters to increase upon 4E-BP-4A expression (Suppl. Fig. 11F). Thus we conclude that neither eIF4E2 nor eIF4E3 is involved in translating resistant mRNAs when eIF4E1 is blocked.

Another protein that has been reported to bind cap is eIF3l[55]. We, therefore, generated *eIF3l* knockout HeLa cells (Suppl. Fig. 12A-A'). Expression of the resistant luciferase reporters still increased upon 4E-BP-4A expression in the *eIF3l* knockout cells (Suppl. Fig. 12B) indicating that *eIF3l* is also not involved in this process.

A previous study identified cap-binding proteins by performing a cap pull-down in the presence or absence of RNase treatment (to identify proteins that bind cap directly) followed by mass spectrometry[47]. From this list, we selected all proteins that bind cap directly and removed ribosomal proteins and known eIFs. In addition, we included members of the LARP family, which are known to have cap-binding properties, resulting in a list of 39 candidates. We then tested each of these candidates by CRISPR/sgRNA-mediated knockout followed by transfection with a resistant reporter (Suppl. Fig. 13A). To

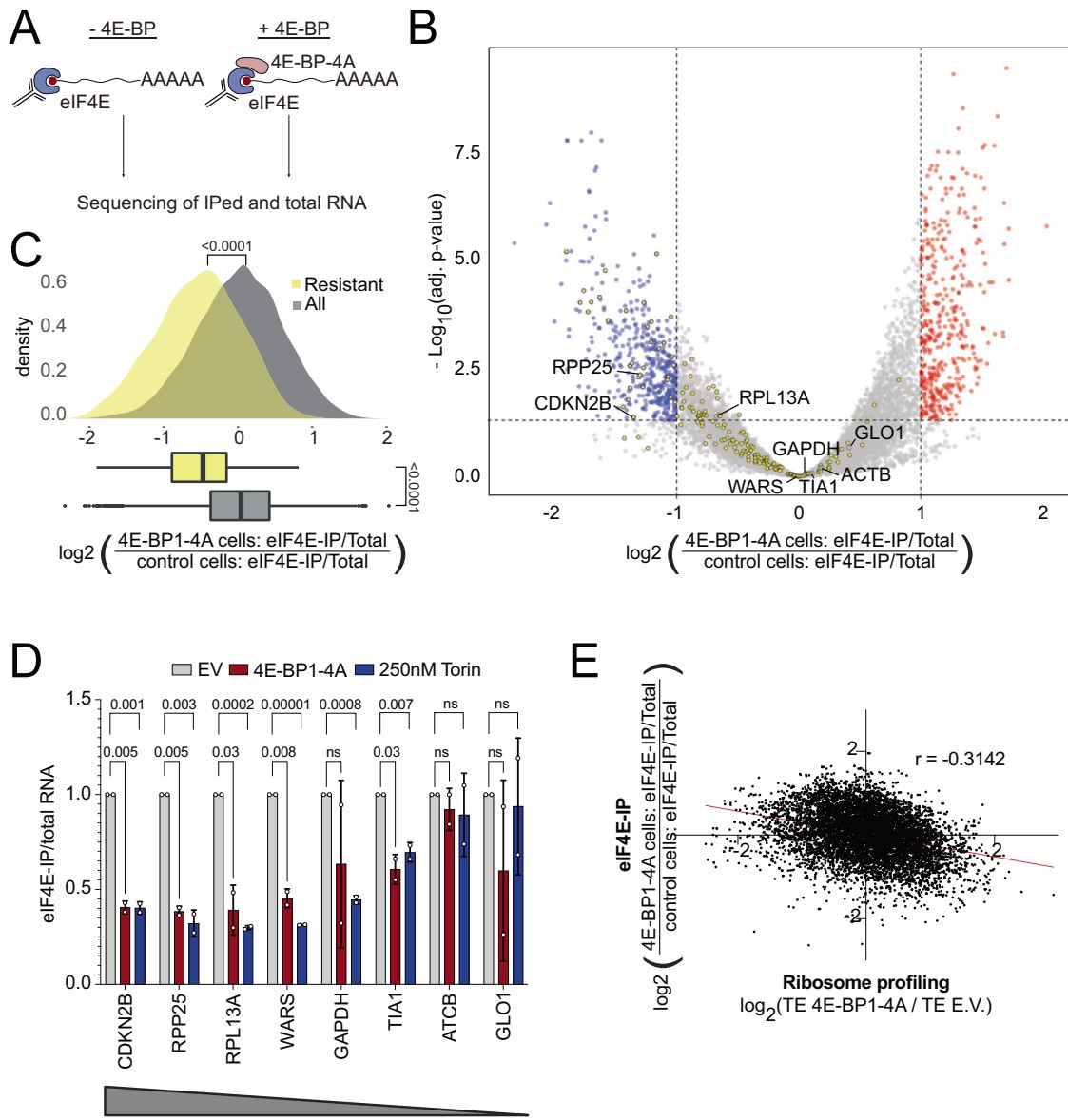

**Fig. 4 | Resistant mRNAs are released from eIF4E upon eIF4E inhibition.**
**A** Schematic illustration of the experiment workflow. **B** Scatter plot of log2(fold change of RNA co-immunoprecipitated with eIF4E, 4E-BP1-4A (full-length)/E.V.) versus significance. Significant candidates with log2(fold change) >1 are red, <−1 are blue. The 183 'resistant' mRNAs obtained from ribosome profiling in Fig. 1A are shown in yellow. Transcripts used for the validation in panel **D** are labeled by gene name. Significance was estimated with the Wald test performed by DESeq2 package, p-values adjusted for multiple comparison. **C** Resistant mRNAs are more strongly released from eIF4E upon eIF4E inhibition than average. Histogram of log2(fold change of RNA co-immunoprecipitated with eIF4E, 4E-BP1-4A(full-length)/E.V.) for resistant mRNAs (yellow) versus all mRNAs (gray). Box plots show

median with lower and upper quartile. Whiskers: full data range. Significance by unpaired, two-sided, t-test, n = 10,528 genes in 2 biological replicates.
**D** Confirmation of RNA-seq readout for eIF4E binding by Q-RT-PCR. mRNAs binding eIF4E were immunoprecipitated from control cells (E.V.), cells expressing 4E-BP1-4A (full-length) or cells treated with Torin (250 nM, 2 h) and quantified by Q-RT-PCR. All values are normalized to total cellular mRNA. error bars=std. dev., significance by Dunnett's multiple comparison test ANOVA. ns=not significant. n = 2 biological replicates and 3 technical replicates. **E** Genome wide anti-correlation resistance to eIF4E inhibition and change in eIF4E binding. r = Pearson correlation coefficient.

simplify the readout, we generated a fluorescent version of the *RPP25* reporter where we placed the *RPP25* 5'UTR upstream of mNeonGreen, and on the same plasmid the control *HBB* 5'UTR upstream of iScarlet (Suppl. Fig. 13B). Analogous to the luciferase version, this fluorescent reporter increases in expression upon 4E-BP-4A expression when assayed by FACS (Suppl. Fig. 13C). We then tested all 39 candidates using this assay, but none completely abolished the ability of the *RPP25* reporter to increase in expression upon 4E-BP-4A expression (Suppl. Fig. 13D and Suppl. Data 7). We retested the 4 knockouts which caused the strongest suppression (*RPA2, FAM120B, NPM1,* and *LARP1*) using

the original luciferase reporter setup and found that knockout of *NPM1* significantly blunts the resistance of the RPP25 reporter (Suppl. Fig. 13E). However, we could not consistently detect significant NPM1 binding on cap pulldowns (Suppl. Fig. 13F). Thus we conclude the effect of NPM1 on the luciferase reporter may be a specific but indirect effect of the known role of NPM1 in ribosome maturation[56].

## eIF4E-independent translation is eIF3d dependent
One protein that has been reported to bind the cap and to promote translation initiation independently of eIF4E is eIF3d[18–23]. We,

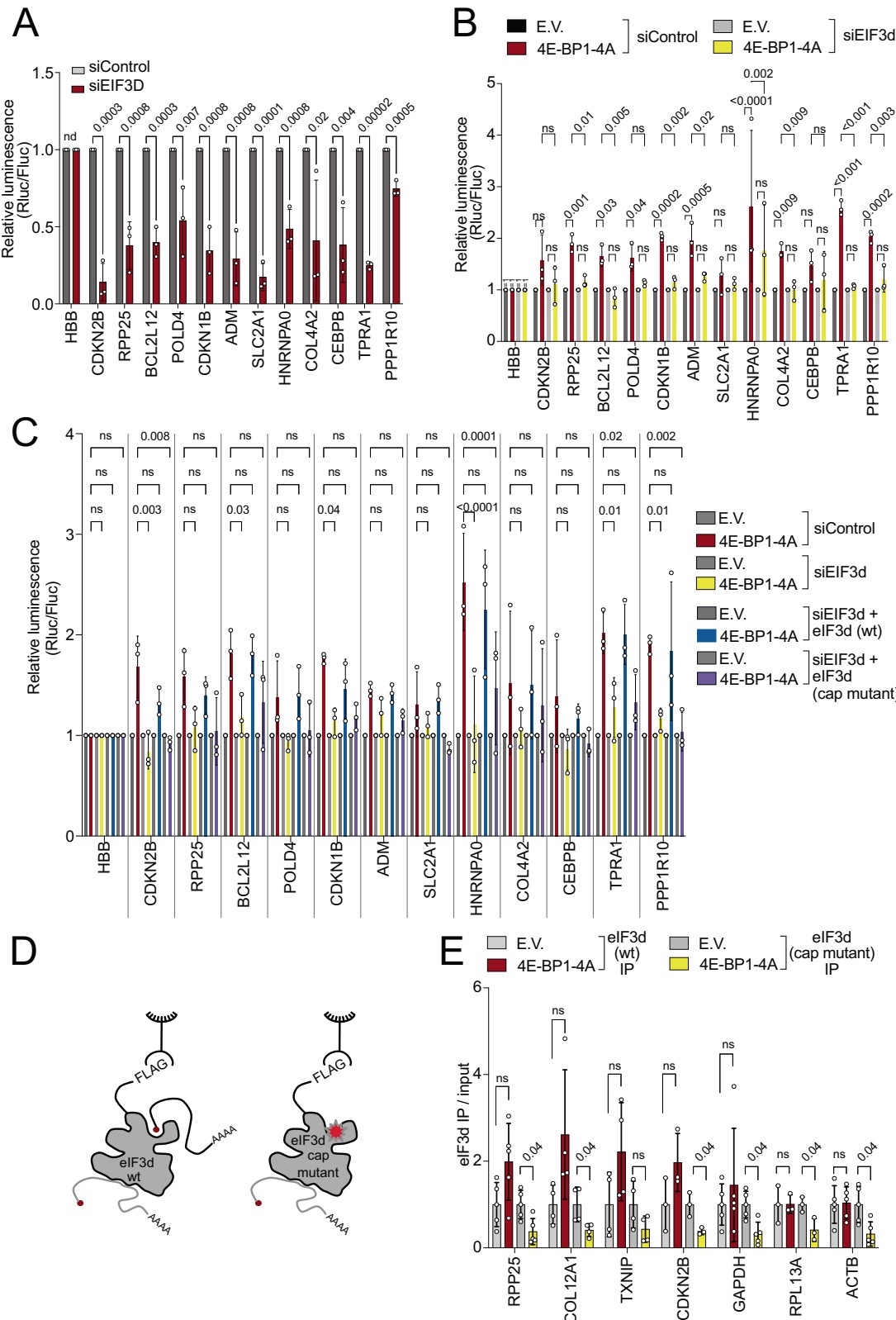

therefore, tested the involvement of eIF3d. Unlike the knockdown of other components of the eIF3 or eIF4F complex, knockdown of *eIF3d* causes a substantial drop in expression of the resistant reporters (Fig. 5A and Suppl. Fig. 14A–C). Note that upon *eIF3d* knockdown there is a general drop in bulk mRNA translation due to the role of eIF3d in the canonical eIF3 complex and in eIF4E-dependent translation[57]. This affects both the resistant RLuc reporters as well as the FLuc

normalization control reporter. However, expression of the resistant RLuc reporters drops more strongly than that of the normalization control, causing a drop in the ratio (Fig. 5A). This indicates that a large fraction of the translation of these reporters depends on eIF3d, even under control conditions when eIF4E is active. We validated these results on endogenous mRNAs by detecting their distribution in polysome gradients from control cells, *eIF3d*-knockdown cells,

**Fig. 5 | eIF4E-independent translation requires cap-mediated binding by eIF3d.**
**A** Translation of resistant reporters is highly dependent on eIF3d. The indicated reporters were transfected into cells that had been treated with siRNAs targeting *eIF3d* or a negative control non-targeting siRNA. **B** The resistance to eIF4E inhibition is strongly blunted upon *eIF3d* knockdown. **C** Resistance to eIF4E inhibition requires the cap-binding capacity of eIF3d. The indicated reporters were transfected into control cells, *eIF3d* knockdown cells, or *eIF3d* knockdown cells reconstituted to express either wildtype or cap-binding-mutant eIF3d. **D, E** Increased

binding of endogenous, resistant mRNAs to eIF3d upon eIF4E inhibition requires the cap-binding capacity of eIF3d. Either wildtype or cap-binding-mutant eIF3d were immunoprecipitated from cells expressing 4E-BP1-4A or empty vector (neg. control) (**D**) and co-IPing mRNAs were quantified by qRT-PCR (**E**). All panels: error bars = std. dev., significance by unpaired, two-sided, t-test adjusted for multiple testing (**A, E**) or by Dunnett's multiple comparison test ANOVA (**B, C**). ns not significant. Panels **A–C** *n* = 3 biological replicates, panel **E**: *n* = 3 for CDKN2B and RPL13A, *n* = 4 for COL12A1, TXNIP, *n* = 5 for the rest of transcripts.

or *eIF3d*-knockdown cells rescued with expression of an *eIF3d* transgene containing silent mutations that evade siRNA targeting (Suppl. Fig. 15). This revealed that *eIF3d* knockdown causes eIF4E-resistant mRNAs such as *RPP25* or *CDKN2B* to shift out of polysomes and into monosomes, but not other mRNAs such as *ACTB* or *RPL13A* (Suppl. Fig. 15B).

We next asked whether the eIF4E-independent translation of these reporters is dependent on eIF3d. Indeed, knockdown of *eIF3d* strongly blunts the resistance of these reporters to 4E-BP-4A expression, causing them to behave similar to the control reporter carrying the *HBB* 5′UTR (yellow bars, Fig. 5B).

Finally, we asked whether the eIF4E-independent translation of these reporters depends on the cap-binding ability of eIF3d. To this end, we knocked-down *eIF3d* and reconstituted the cells with either wildtype eIF3d or a cap-binding mutant eIF3d(D249Q/V262I/Y263A)[19]. This comparison between wildtype and mutant eIF3d is critical because eIF3d functions both as a component of the canonical eIF3 complex which promotes eIF4E-dependent translation, and as a component of this non-canonical initiation mechanism which binds cap directly. Hence knockdown of *eIF3d* removes both the canonical and cap-binding-mediated functions of eIF3d. Indeed, reconstitution of the cells with cap-binding mutant eIF3d restores the majority of translation in a cell, similar to wildtype eIF3d, as detected by bulk incorporation of AHA (Suppl. Fig. 16A, B) or by polysome gradients (Suppl. Fig. 16C). However, in cells reconstituted with cap-binding mutant eIF3d, but not wildtype eIF3d, the resistant reporters lost their resistance and could no longer be highly translated when 4E-BP-4A is expressed (Fig. 5C). Thus, eIF4E-independent translation specifically requires the cap-binding ability of eIF3d.

eIF3d has been reported to work together with eIF4G2/NAT1 to promote selective translation of mRNAs encoding epithelial-to-mesenchymal (EMT) regulators[18,22,23]. Thus, we tested if eIF4G2 is also involved in enabling the resistant reporters to evade suppression upon eIF4E inhibition. Indeed, *eIF4G2* knockdown partially reduced the ability of the resistant reporters to be translated upon 4E-BP-4A expression (Suppl. Fig. 16D), in agreement with eIF3d and eIF4G2/NAT1 working together.

### Resistant mRNAs shift from eIF4E binding to eIF3d binding upon eIF4E inhibition

We previously found that resistant mRNAs preferentially detach from eIF4E when eIF4E is inhibited (Fig. 4) and hypothesized this must be due to their binding to another cap-binding protein. To test whether they bind eIF3d, we performed eIF3d pulldowns from control cells or cells expressing 4E-BP-4A and detected endogenous mRNAs by Q-RT-PCR. This revealed that binding of the resistant mRNAs such as *RPP25* or *COL12A1* to wildtype eIF3d increases when eIF4E is inhibited (red bars, Fig. 5D, E). In contrast, binding to cap-binding mutant eIF3d does not increase (yellow bars, Fig. 5D, E) indicating that the cap-binding ability of eIF3d is required for these resistant mRNAs to shift from eIF4E binding to eIF3d binding when eIF4E is inhibited. In sum, our data support a model whereby upon eIF4E inhibition, a set of mRNAs shift from binding eIF4E via their caps to binding eIF3d via their caps, and this enables them to continue being translated.

### eIF4E-resistant mRNAs have eIF3 binding sites in their 5′UTRs

What causes some, but not all, mRNAs to be translated in an eIF3d-dependent manner? Since eIF3d is thought to bind the standard cap, which is present on most cellular mRNAs, this is unlikely to be the discriminatory factor. To address this, we focused on two of the resistant reporters that show the strongest and most consistent response to eIF4E inhibition, the *RPP25* and *CDKN1B* reporters. We performed serial truncations of both the 5′- and 3′- ends of their 5′UTRs to identify if there are any elements which impart the resistance. This revealed that indeed the *RPP25* 5′UTR has a 50nt region between nt 100-149 that is required for the resistance (Suppl. Fig. 17A). Likewise, the *CDKN1B* reporter requires nt 150-272 for resistance (Suppl. Fig. 17B). To validate the truncation results, we deleted nt 90–140 from the *RPP25* reporter and found that this blunts the resistance (Suppl. Fig. 17C). Conversely, introducing nt 100–150 of the *RPP25* 5′UTR into the middle of the negative control *HBB* reporter is sufficient to make it fully eIF4E-resistant (Suppl. Fig. 17C) indicating that these 50nt contain the responsible functional element. Likewise, introducing nt 150-272 from the *CDKN1B* reporter into the HBB 5′UTR also induces resistance (Suppl. Fig. 17D). We asked what these two regions have in common? We did not find any motif or secondary structure in common between the two regions, however, both regions were identified as eIF3a, eIF3b, and eIF3d binding regions in CLIP experiments (Suppl. Fig. 17E) (POSTAR3 database [58,20]). This suggests that eIF3d requires additional sequences within a 5′UTR to efficiently bind the cap of an mRNA. Indeed, binding of RLuc reporters carrying the *RPP25* or *CDKN1B* 5′UTRs to eIF3d was reduced when we deleted the respective functional elements (Suppl. Fig. 17F).

### The eIF3d functional element must be <400 nt from the cap

We next asked if there are any requirements regarding the positioning of the functional element. To this end, we started with a luciferase reporter containing the *HBB* 5′UTR, into which we introduced the functional element of *CDKN1B* (Suppl. Fig. 18A). We then either increased the distance between the functional element and the cap by multimerizing the intervening sequence, or we reduced the distance by performing truncations. We then tested the ability of these reporters to be expressed in the presence of 4E-BP-4AΔTOS. This revealed that the reporter resistance was impaired when the distance of the functional element to the cap was increased beyond 300nt (Suppl. Fig. 18B) but there was no requirement for a minimal distance to the cap (Suppl. Fig. 18C). Likewise, we either increased or decreased the distance between the functional element and the RLuc start codon, but this had no effect (Suppl. Fig. 18D, E). In sum, the functional element cannot be too far from the cap, which fits with the concept that this element enables eIF3d to 'land' on the 5′UTR to bind the cap.

### Secondary structure of the functional element

We next sought to understand if the secondary structure of the functional element is important. To that end, we first narrowed down further the key sequences within the functional element. We mutated blocks of 10 consecutive nucleotides to tile the entire 108 nt long CDKN1B functional element (Suppl. Fig. 19A). Since the functional element has few adenosines, we mutated in each case the 10 nucleotides to adenosine. This identified nucleotides 214-223 (TGGTCCCCTC) as being important (Suppl. Fig. 19B). We then

performed a scanning mutagenesis of this sequence in blocks of 2 nucleotides at a time, but this did not reveal any specific, critical residues (Suppl. Fig. 19C).

To study the secondary structure of this element, we performed SHAPE analysis, which is based on the reactivity of 2'OH groups of unpaired nucleotides, using the entire *CDKN1B* 5'UTR (Suppl. Fig. 19D). This revealed that the secondary structure is unlikely to be important, since the functional element (nt. 164-272) does not fold within itself, but pairs with sequence further downstream in the *CDKN1B* 5'UTR which is not present when the functional element is isolated and cloned into the *HBB* 5'UTR (Suppl. Fig. 19A). In particular, the 10 nucleotides identified above (214–223) pair with nt 381–390. To test if this pairing is important, we took the RLuc reporter carrying the *CDKN1B* full-length 5'UTR and mutagenized resides 219-222 (CCCC) of the functional element, and the corresponding residues on the other side (383–386, GGGG), so that they either pair or do not pair, but this had no effect on the response to 4E-BP-4A (Suppl. Fig. 19E). In sum, the secondary structure of the *CDKN1B* functional element does not seem to be important.

Surprisingly, at the primary sequence level, the functional element appears to be made of multiple redundant pieces. For instance, the truncations in Suppl. Fig. 17A, B do not give an all-or-nothing response - some truncations retain a partial response to 4E-BP-4A, suggesting that some responsive sequence was removed, and some was retained. Furthermore, we did not find the TGGTCCCCTC sequence from *CDKN1B* in the *RPP25* functional element, suggesting a different sequence may be key there. Instead, the *RPP25* functional element is G/C-rich, reminiscent of eIF4G2 binding sites[59]. Further work will be needed to better understand the sequence requirements of these functional elements.

### The eIF3d binding sequence potentiates translation also in non-stressed conditions

We were intrigued by the fact that the reporters carrying the 5'UTRs of '4E-BP resistant genes' were not only sensitive to *eIF3d* knockdown in the 4E-BP-4A expression condition, but also in the control, non-stressed condition (Fig. 5A). One possibility is that also in control conditions endogenous 4E-BP is slightly active and therefore the cap-binding capacity of eIF3d promotes translation on these mRNAs. Another possibility is that the functional element we identified above helps to recruit the eIF3 complex, and hence to promote translation, also in non-stressed conditions when eIF4E is active and binding the cap. To distinguish between these two options, we tested whether the cap-binding activity of eIF3d is required to promote translation of the resistant genes in non-stress conditions. Interestingly, both wildtype eIF3d as well as cap-binding-deficient eIF3d were able to rescue the drop in translation caused by *eIF3d* knockdown on reporters containing the 5'UTRs of resistant genes (Fig. 6A). This indicates that in non-stressed conditions eIF3d promotes translation on these 5'UTRs independently of its cap-binding capacity. Indeed, multimerization of the *CDKN1B* functional element in the heterologous context of the HBB 5'UTR led to a massive 50-fold induction in reporter expression (Fig. 6B). This did not occur if the CMV promoter was absent from the reporter, indicating that the 108nt *CDKN1B* functional element does not code for a cryptic promoter (Fig. 6B). Normalization of luciferase activity levels to reporter mRNA levels confirmed this is a translational effect (Fig. 6C). Multimerization of the functional element caused the reporter to become more dependent on eIF3d for its expression (Fig. 6D) indicating that the activity of the functional element is eIF3d-dependent.

### Discussion

The canonical and predominant mode of translation initiation in unstressed human cells in culture is directed by eIF4E1 upon binding the cap[3]. In addition, several other non-canonical translation initiation

mechanisms exist, including m6A-dependent initiation[15,16], eIF4E2-dependent initiation[12,13], IRES-dependent initiation, and eIF3d-dependent initiation[18–20]. Our data suggest that when eIF4E is inhibited, the main alternate mode of translation initiation is the one depending on the cap-binding activity of eIF3d. The other mechanisms are likely predominant on specific mRNAs or under particular conditions such as hypoxia[12,13] or heat shock[15,42,43]. Our data suggest a model whereby some mRNAs have an eIF3d binding sequence which acts to recruit eIF3d to the mRNA (Suppl. Fig. 17F). In non-stressed conditions, this potentiates translation of the mRNA in a manner that is independent of cap-binding by eIF3d, perhaps via canonical eIF4E-dependent translation (Fig. 7). Instead, when eIF4E is inactivated, the cap-binding capacity of eIF3d comes into play, enabling these mRNAs to continue being translated, while mRNAs lacking this eIF3d binding site become less translated (Fig. 7). Thus the eIF3d-binding sequence potentiates translation both when eIF4E is active and inactive, although the distance to the start codon is not relevant in the case of the eIF3d sequence.

Our data are consistent with several publications. The Schneider lab found that under normal physiological conditions roughly 20% of mRNAs are translated in an eIF3d-dependent manner[22]. A recent publication employed a proteomic analysis to find that proteins involved in cell adhesion and migration are still synthesized when mTORC1 is inhibited, that this is blunted when *eIF3d* is knocked down, and that this correlates with increased binding of their mRNAs to eIF3d[60]. Dendrite pruning in Drosophila neurons has also been shown to depend on activation of 4E-BP1 and on the cap binding capacity of eIF3d[61]. Likewise, translation in human cells infected with human cytomegalovirus becomes progressively less dependent on eIF4E and more dependent on eIF3d[62].

Our eIF4E-IP and eIF3d-IP data (Figs. 4B and 5E) suggest that a subset of mRNAs reduce their binding to eIF4E and instead increase their binding to eIF3d when 4E-BP-4A is expressed. This suggests there may be a competition in cap binding between these two factors, with eIF4E predominating when it is active. This also suggests that 4E-BP binding to eIF4E can cause eIF4E to detach from mRNA. Although initial data indicated that 4E-BP1 association with eIF4E increases eIF4E affinity for the cap[63], our data are more consistent with recent data showing that 4E-BP1 binding to eIF4E leads to its dissociation from mRNA[64] and to older biophysical assays[65]. Considering that the eIF4E-4E-BP1 complex was reported to relocate to the nucleolus[66], it would be surprising if it does so in the context of a whole RNP complex.

When eIF4E is inhibited, binding of eIF3d to some mRNAs such as *RPP25* increases, while binding to others such as *RPL13A* or *ACTB* does not (Fig. 5E). What causes this selective binding of eIF3d to some mRNAs? By bashing the 5'UTRs of *RPP25* and *CDKN1B* we identified small regions of circa 50nt that are required and sufficient to impart eIF4E-resistance to luciferase reporter constructs (Suppl. Fig. 17), suggesting that these regions contain important regulatory information. Indeed, CLIP data indicate that these regions bind the eIF3 complex (Suppl. Fig. 17E). Thus, we think it is likely that these regions contain sequences that are recognized by the eIF3 complex which are needed in combination with cap binding to increase mRNA binding recognition and affinity. We did not notice any primary sequence motifs or secondary structures that are similar in the two regions, however identification of more such regions from a handful of additional mRNAs may be required to add statistical power to such searches.

### Methods
#### Cell lines and culture conditions
All cell lines were cultured in DMEM (Gibco 41965039) + 10% fetal bovine serum (FBS) (Sigma, S0615) + 100 U/ml Penicillin/Streptomycin (Gibco 15140122). To assay de novo protein synthesis, cells were transiently incubated in Methionine-free RPMI (Gibco A1451701) +

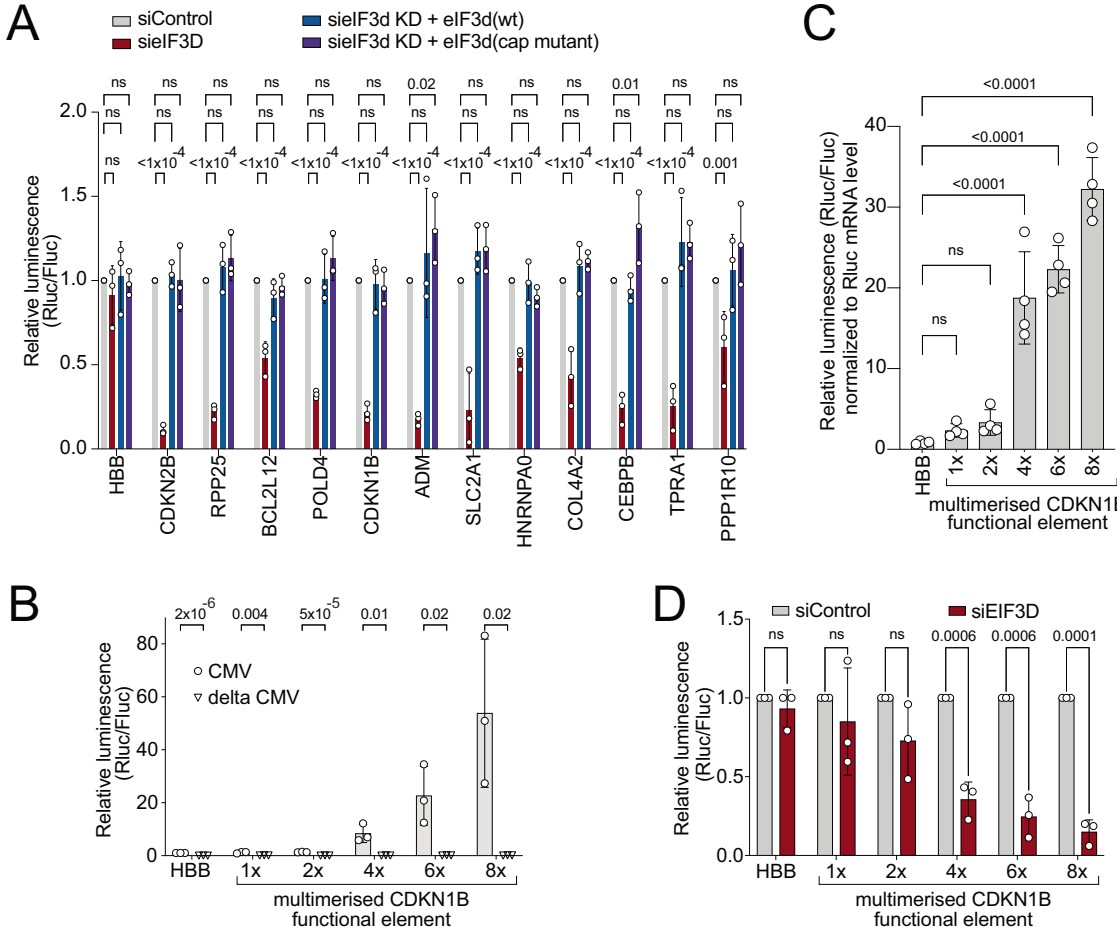

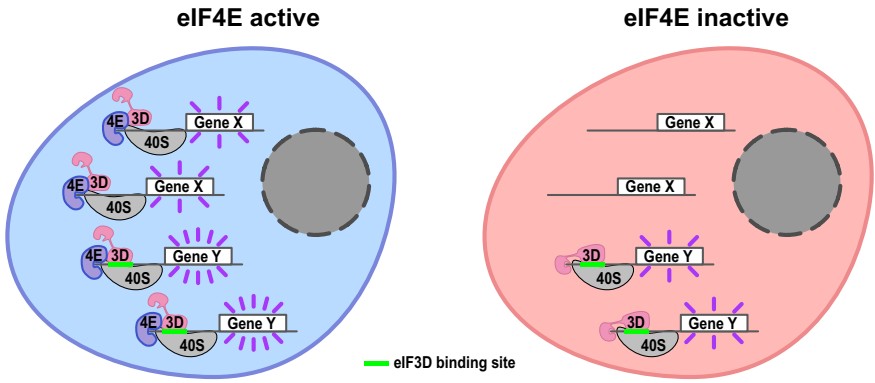

**Fig. 6 | The eIF3d binding element potentiates translation also in non-stressed conditions in a cap-binding-independent manner. A** eIF3d is needed for expression of the resistant reporters in non-stressed conditions independently of its cap-binding activity. The indicated reporters were transfected into control cells, *eIF3d* knockdown cells, or *eIF3d* knockdown cells reconstituted to express either wildtype or cap-binding-mutant eIF3d. **B**, **C** Multimerization of the *CDKN1B* functional element strongly boosts reporter expression in non-stressed conditions. Either 1, 2, 4, 6, or 8 tandem copies of the 108nt long *CDKN1B* functional element were cloned into the middle of the *HBB* 5′UTR of an RLuc reporter. **B** Luciferase activity was measured for plasmids either containing a CMV promoter to drive expression of the reporter ("CMV") or lacking a CMV promoter ("delta CMV") to control for the presence of a cryptic promoter in the *CDKN1B* functional element. **C** Luciferase activity levels were normalized to reporter mRNA levels, quantified by qRT-PCR, to confirmation it is a translational effect. **D** Multimerization of the *CDKN1B* functional element renders a reporter more dependent on eIF3d for its expression in non-stressed conditions. The same reporters as in panel **B** were transfected into control cells or *eIF3d* knockdown cells. All panels: data are the mean values, error bars = std. dev, significance by unpaired, two-sided, *t*-test adjusted for multiple testing (**B**, **D**) or by Dunnett's multiple comparison test ANOVA (**A**, **C**). ns not significant. *n* = 3 for panels **A**, **B**, and **D**, *n* = 4 for panel **C**.

**Fig. 7 | Schematic illustration of proposed model.** The data are consistent with a model whereby a subset of mRNAs shifts upon eIF4E inhibition from binding eIF4E via their caps to binding eIF3d via the cap, and this enables them to continue being translated. This depends on the presence of an eIF3d binding site in the 5′UTR. This site also helps recruit eIF3d and potentiate mRNA translation in non-stressed cells, although in this case the cap-binding capacity of eIF3d is not needed.

dialyzed serum (Life Technologies, A33820-01) supplemented with either Methionine (Sigma, M53098) or L-azidohomoalanine (Sigma, 900892). Cell splitting was performed using Trypsin-EDTA (Gibco, 25200056). All cell lines were tested negative for mycoplasma and authenticated using SNP typing.

## Generation of knockout cell lines and targeted CRISPR-Cas9 screen

All knock-out cell lines were generated using CRISPR-Cas9 technology. sgRNA sequences were obtained from the Brunello library[67]. Sequences of sgRNAs are listed in Suppl. Data 1. To generate the sgRNA plasmids, oligos with the sgRNA sequences were cloned into pX459V2.0[68] via the Bbs1 site. Wildtype HeLa cells were transfected with the sgRNA plasmids using Lipofectamine 2000 in a ratio 2:1 reagent:DNA (Life Technologies, 11668500). 24 h post-transfection, the medium was replaced with a medium containing 1.5 µg/ml puromycin (Sigma-Aldrich, P9620). Three days after puromycin selection, surviving cells were shifted into normal medium (1x DMEM, 10% fetal bovine serum, 1% Penicillin/Streptomycin) and cultured to reach confluency. To generate knockout clones, surviving cells were subjected to single clone selection by serial dilution into 96-well plates. Expanded single clones were tested for the loss of the protein of interest by immunoblotting. All knock-outs were validated by genotyping. For targeted CRISPR-Cas9 screens, cells were re-seeded either in 24-well format at a seeding density of 40.000 cells/well in 500 µl of medium or in 96-well plates at a seeding density of 8000 cells/well. The day after seeding, cells were transfected with reporters by use of Lipofectamine 2000 transfection reagent and 24 h post transfection the activity of reporters was recorded.

## Generation of cell lines stably expressing proteins

To generate cell lines that stably express the variants of *eIF3d*, the corresponding eIF3d constructs were cloned into a piggyback plasmid (pMR368), yielding pMR914 (siRNA-resistant, WT eIF3d) or pMR915 (siRNA-resistant, eIF3d cap-binding mutant). The plasmid of interest and transposase were transfected into HeLa cells in the ratio 1:1. 24 h post transfection, medium was exchanged with medium containing 1.5 µg/ml puromycin (Sigma, P9620). Cells were selected for three days. The surviving cells were shifted into normal medium (1x DMEM, 10% fetal bovine serum, 1% Penicillin/Streptomycin) and cultured to reach confluency. Single clone selection was performed by serial dilution into 96-well plates. The screening of positive clones was performed by western blot. As a result, the HeLa cell lines constitutively expressing eIF3d variants are resistant to puromycin.

## Preparation of cell lysates with RIPA buffer

Cells were seeded at a density of either 500.000 cells per 6-well or 1.500.000 cells per 10 cm dish. The following day, medium was removed and cells were briefly washed with PBS. All residual PBS was removed from the dish and cells were lysed with 150-300 µl of RIPA buffer supplemented with 20U of Benzonase (Merk Millipore, 70746-3), protease (Sigma, 4693159001) and phosphatase (Sigma, 4906837001) inhibitors. Cells were collected by scraping and the lysate was clarified by centrifugation at 4 °C for 10 min at 20,000 g. The protein concentration was measured by Pierce BCA (Life technologies, 23224, 23228). Based on the observed concentration, samples were balanced to equal protein concentration and mixed with 5x Laemmli buffer(1/5 of the final volume). Samples were incubated at 95 °C for 5 min and then loaded on SDS gel.

## Western blotting

Cell lysates were run on SDS gels, and transferred to a nitrocellulose membrane with 0.2 or 0.4 µm pore size (Amersham, 10600001 or 10600002) by wet transfer. The quality of transfer was estimated by ponceau-S (Serva, 33427.01) staining. The membranes were blocked in 5% skim milk / PBST for 1 h, followed by overnight incubation in primary antibody solution (5% BSA / PBST) at 4 °C. After overnight incubation, membranes were washed three times in PBST to remove unbound primary antibodies followed by incubation in secondary antibody (1:10,000 in 5% skim milk / PBST) for 2 h at room temperature. Prior to development, membranes were washed three times for 15 min in PBST. Finally, chemiluminescence was detected with ECL reagents (Thermo Schientific, 32109) and imaged with a Biorad ChemiDoc imaging system. Antibodies used for immunoblotting are listed in Suppl. Data 2.

## Cloning

Firefly (pAT1620) and Renilla (pAT1618) luciferase reporters used to introduce 5'UTRs of interest were previously generated in ref. 69. The 5'-UTRs of interest were amplified from cDNA of wildtype HeLa cells and cloned into pAT1618 between the HindIII and Bsp119l sites. The normalization control HBB 5' UTR was cloned via HindIII and NheI sites into pAT1620. In case the 5'-UTR was short enough, it was introduced directly by oligo cloning. All oligos used for cloning 5' UTRs are listed in Suppl. Data 3. All cloned 5'UTRs variants are listed in Suppl. Data 4, with indication of their transcript IDs. To prepare the template for in vitro transcription, a plasmid containing a T7 promoter, Renilla luciferase, and a synthetic 73-nucleotide poly-A track was generated (pMR172). To clone the 5'-UTRs of interest into pMR172, the 5'-UTRs were amplified with oligos carrying Pst1 and Bst119I sites and then cloned between these sites into pMR172. The bi-cistronic plasmid containing an upstream RLuc and downstream FLuc was a gift by Dr. Katharina Clemm von Hohenberg. To clone the 5'-UTRs of interest into this bi-cistronic construct, the 5'-UTRs were amplified with oligos caring EcoR1-Eco32I sites and cloned via these sites into the inter-cistronic region. A version of the bi-cistronic reporter suitable for in vitro transcription was generated by re-cloning the Rluc-5'-UTR-Fluc part into pMR172 via Bsp119I-XbaI sites, resulting in plasmids where Rluc is preceded by the HBB 5' UTR and Fluc by the 5'-UTR of interest. For plasmids with a stable stem-loop at the beginning of the 5'UTR, the sequence GGGAGTGGACTTCGGTCCACTCCC, containing a BoxI site was introduced into the constructs via inverted PCR, which allowed to place 5'-UTRs of interest directly after the stem-loop. The 5' UTRs of interest were subcloned into the resulting construct via BoxI-Bsp119I sites. Shortening of the *RPP25* and *CDKN1B* 5'-UTR was achieved by inverse PCR with omission of the fragments requiring deletion, and plasmid re-ligation. Generation of chimeras of *HBB-RPP25* or *HBB-CDKN1B* was achieved by the introduction of Kpn2I BshT1 sites into the 5'-UTR of *HBB*. The regions of interest in *RPP25* and *CDKN1B* were PCR amplified with oligos caring Kpn2I and BshT1 sites and cloned via these sites into the *HBB* 5'-UTR. To perform the tiling mutagenesis of the *CDKN1B* functional element, an oligo cloning approach was used. Oligos listed in Suppl. Data 3 were cloned into the *HBB* 5'UTR via Kpn2I and BshT1. To change the distance of the *CDKN1B* functional element to the cap or to the Rluc start codon, oligos of different lengths listed in Suppl. Data 3 were cloned via the HindIII and Kpn2I sites as follows: via HindIII and Kpn2I (to move it closer to the cap), HindIII site (to move it further from the cap), BshT1 and Bsp199I sites (to move it closer to Rluc AUG) or BshT1 site (to move it further from AUG). To generate in vivo uncapped reporter, the U6 promoter was cloned upstream of the *RPP25* 5' UTR – Rluc. To omit PolII premature termination (presence of four T in row), the Rluc coding sequence was optimized by introducing synonymous mutations that disrupts the repeats of four T in a row. The CTE export element was first assembled by oligos in the TOPO vector (Life Technologies, 450245) and then cloned into the final construct via XbaI-ApaI sites. A PolII termination signal was introduced by inverted PCR. *eIF3d* mutants were generated by site-directed mutagenesis and cloned into the piggy back backbone via Bspt1 – Not1 sites. The *4E-BP1-4A* construct was a gift of Dr. Gianluca Figlia. The TOS-motif was removed from this plasmid by inverted PCR. The PEST

domain was introduced into the NeonGreen and iScarlet constructs by PCR with the oligos listed in Suppl. Data 3. All generated constructs were verified by sanger sequencing. All plasmids are available at the European Plasmid Repository.

### Deproteinization of template for the in vitro transcription

Plasmids used as a template for in vitro transcription were linearized by HindIII, which is placed directly downstream of the synthetic poly(A) tail encoded in the plasmid. In the case of EMCV-bearing constructs, the plasmid was linearized with BoxI or Eco47III, due to presence of a HindIII site in the 5′-UTR of EMCV. The linearity of the plasmid was checked by running an aliquot of digestion mix on an agarose gel. Proteinase K (final concentration 100 μg/ml) and SDS (final concentration 0.5%) were then added to the digested plasmid. The mix was incubated with shaking at 50 C for 30 min, followed by extraction with Phenol-Chloroform-Isoamylalcohol (detailed below).

### In vitro transcription of RNA, RNA capping, and SHAPE

In vitro transcription was performed using the MegaScript T7 transcription kit according to manufacturer's recommendation with some modifications mentioned below (Megascript AM1334). To obtain the m7G- (NEB, S1411L) and A-capped (NEB, S1406L) transcripts, co-transcriptional capping was performed. For this, the A- or conventional cap was added to the reaction mix to the level of other nucleotides (final concentration 7.5 mM) and the concentration of GTP in the in vitro transcription reaction was lowered by 5-fold (final concentration 1.5 mM). In vitro transcription was performed overnight at 37 C with subsequent purification of RNA by phenol-chloroform isoamyl alcohol extraction (detailed below). The integrity of RNA was checked by denaturing agarose electrophoresis. When A-capped transcripts were not required, in vitro transcription was performed conventionally according to Megascript kit's recommendations and the synthesized RNA was capped and 2′-O-methylated with the use of the Vaccinia capping system (NEB, M2080S) 2′-O-methyltransferase (NEB, M0366S) following the supplier's protocol. The SHAPE reactivity of capped RNA bearing *CDKN1B* 5′UTR and Rluc ORF was determent by Eclipsbio. The obtained SHAPE reactivity was used to predict secondary structure with the use of RNAStructure[70].

### Phenol-chloroform isoamyl alcohol extraction of DNA or RNA

The volume of the sample was increased to 300 μl by adding RNase-DNase-free water and then 300 μl of Phenol-Chloroform isoamyl alcohol was added. The mix was shaken vigorously, followed by centrifugation at room temperature for 2 min at maximum speed. The supernatant was transferred into a new tube with 300 μl of Chloroform, followed by vortexing and centrifugation at room temperature for 2 min at maximum speed. Finally, the supernatant was moved into a new tube containing the same volume of isopropanol, 2 μl of Glycoblue (Invitrogene AM9516) and 1/10 volume of 3 M NaAc pH5.2. The sample was incubated overnight at −80C with subsequent centrifugation at 4 C for 30 min at 14000 rpm and a quick 70% ethanol wash. The DNA or RNA-pellet was resuspended in RNase- DNase-free water.

### Acid phenol extraction of RNA

For acid phenol extraction, the sample volume was adjusted to 700 μl with 10 mM Tris pH 7.0. 750 μl of prewarmed acid phenol was added to the sample, followed by incubation of the mix at 65 °C with shaking at 1400 rpm for 15 min. Then samples were chilled on ice for 5 min with subsequent centrifugation at 20,000×*g* for 2 min. The supernatant phase was transferred into a new tube and mixed with 700 μl of acid phenol, followed by incubation at room temperature for 5 min. The sample was then spun at 20000 g for 2 min and the supernatant was mixed with 600 μl of chloroform by vortexing. Finally, this mix was centrifuged at 20,000×*g* for 2 min and the supernatant was mixed with the same volume of isopropanol, 2 μl of Glycoblue (Invitrogene AM9516) and 1/10 volume 3 M NaAc pH5.2. The sample was incubated overnight at −80C with subsequent centrifugation at 4 C for 30 min at 14000 rpm and a quick 70% Ethanol wash. The DNA or RNA-pellet was resuspended in RNase- DNase-free water.

### Ribosome profiling

HeLa cells were seeded at 1.5 million cells per 15 cm dish in 20 ml of growth medium two days before harvesting. The following day, cells were transfected either with 4E-BP1-4A or empty vector constructs. 24 h post transfection, cells were harvested in the following way: the growth medium was quickly poured off and cells were quickly washed with ice-cold washing solution (1x PBS 10 mM MgCl2, 800 μM Cyclo-heximide). The wash solution was poured off and all residual wash solution was removed by gentle taping of the 15 cm dish on its side for a few seconds. The cells were lysed with the addition of 150 μl of lysis buffer (0,25 M HEPES pH 7.5, 50 mM MgCl2, 1 M KCl, 5% NP40, 1000 μM Cycloheximide). Cells were scraped into an Eppendorf tube and the lysate was clarified by centrifugation at 20,000×*g* for 10 min at 4 °C. The concentration of lysate was estimated using a nanodrop spectrophotometer, measuring RNA content against a water-blanked control. A fraction of lysate was used for total RNA preparation (200 μl) and the rest was used for footprint generation. To digest polysomes into monosomes, 40S and 60S lysates were incubated with RNase I (100 Unit per 120 μg of lysate) on ice for 30 min. Digested lysates were loaded on the top of a 15-65% sucrose gradient, which was prepared with the use of a Biocomp Gradient Master. To separate 80Ss from cytosolic fractions, 40Ss, and 60Ss the gradient was ultracentrifuged for 3 h at 35,000 rpm in a Beckman Ultracentrifuge with a SW40 rotor. The gradient was subsequently fractionated on a Biocomp Gradient Profiler system and 80S-containing fractions were collected. RNA from these fractions was extracted with the acid-phenol RNA extraction protocol described above. The integrity of RNA was analyzed on a Bioanalyser, with the use of the total RNA Nano 6000 Chip. If RNA passed the quality test, the selection of footprints was performed. For this, extracted RNA was run on a 15% Urea-Polyacrylamide gel and fragments of 25–35 nucleotides were purified from the gel. The gel pieces containing the footprints were smashed into small pieces with gel smasher tubes. 0.5 ml of 10 mM Tris pH 7 were added to the smashed gel pieces and the suspension was incubated at 70 °C for 10 min with shaking. The mix was briefly centrifuged and the supernatant was used for RNA precipitation by isopropanol. Purified footprints were dephosphorylated by means of T4 PNK (NEB) for 2 h at 37 °C in PNK buffer without ATP. After this, the footprints were again precipitated and purified using isopropanol. Finally, footprint quality was assayed with an Agilent Bioanalyzer small RNA chip and Qubit smRNA kit. The libraries were prepared strictly in accordance with the Next-Flex small RNA v.4 kit protocol (Perkin Elmer, NOVA-5132-06). Total RNA libraries were prepared using the Illumina TruSeq Stranded library preparation kit. The libraries were sequenced on an Illumina Next-Seq 550 system.

### Data analyses of ribosome profiling

The sequences of adapters and randomized nucleotides derived from the use of the Nextflex kit were removed with cutadapt software. Then the reads aligning to tRNA or rRNA were removed using bowtie2. All remaining reads were mapped to the human transcriptome (Ensemble transcript assembly 94) and genome (hg38) using BBMap software, with multiple mapping allowed. Transcripts with PCR artefacts were removed from the analysis and are listed in Suppl. Data 8. The reads mapping to the coding sequences were quantified with lab-based software written in C. Read counts for each transcript were analyzed using DESeq2 software to obtain values of translation efficiency, log2 fold changes, and adjusted p-values. Motif analysis search was performed using MEME software (parametersL -minW 6 – maxW 10, -maxsize 10000000 -dna -nmotifs5 -maxsites 200). GSEA analysis was performed using Web-Base Gene SeT AnaLysis Toolkit (Webgestalt).

### De novo protein synthesis assay (BONCAT)

On the day prior to the assay, cells were seeded in 15 cm dishes to reach 80% confluence on the day of the experiment. 30 min prior to labeling, cells were washed with PBS and shifted into Methionine-free RPMI, 10% FBS, 1% Pen-Strep medium, followed by addition of 0.8 mM AHA (Sigma, 900892) or methionine (Sigma, M5308). The negative control with cycloheximide was performed by co-incubation of cells in 0.8 mM AHA and 100 µg/ml cycloheximide. Cells were labeled for 2 h. After labeling, cells were quickly washed in PBS and lysed with 500 µl of Click-IT Chemistry Lysis Buffer (115 mM TrisHCl pH8,5, 1% NP40, 1x inhibitors of proteases, 20U of benzonaze). The lysates were clarified by centrifugation and used directly in a Click-IT reaction. For this, the lysate was mixed with CuSO4 (final. 1.9 mM), ascorbic acid (final 1.9 mg/ml), Acetylene-PEG Biotin (final 2.2 nM) and incubated for 30 min at room temperature rotating. After the Click-IT reaction, excessive biotin was removed by methanol precipitation. For this 3 sample volumes of methanol were added to the sample, followed by vortexing. Then 0.75 sample volumes of chloroform were added and vortexed. Finally 2 sample volumes of water were added and vortexed. This mix was centrifuged at room temperature at 14,000 rpm for 2 min and the upper phase was discarded, without perturbing the interphase. To the remaining sample, 2 volumes of methanol were added followed by vortexing and centrifugation at room temperature at 14000 rpm for 2 min. The pellet was air-dried for 10 min and then resuspended in 120 µl of Click-it resuspension buffer (1% SDS, 1% NP40, PBS 1x, 1x protease inhibitors). The resuspended sample was quickly centrifuged and transferred into the new tube, to remove any undissolved material. The volume of the sample was adjusted to 2 ml by Neutravidin binding buffer (0.1% SDS, 1% NP40 1x PBS, 1x protease inhibitors). The protein concentration of the samples was estimated by BCA and the samples were equalized based on the observed values. Small aliquots were set apart to serve as an input control, while the rest of the sample was incubated with 100 µl of Neutravidin agarose (Thermofisher Scientific 29201) overnight, at 4 C rotating. The following day, a series of washes were performed. First the beads were washed once with Neutravidin binding buffer (0.1% SDS, 1% NP40 1x PBS, 1x protease inhibitors), followed with two 15 min washes with 4 M Urea, 1x PBS, 1x protease inhibitors and two 15 min washes with 6 M Urea, 1x PBS, 1x protease inhibitors. At the final wash, the beads were transferred into a new tube. Elution was performed by incubation of the beads with 50 µl of 2x Laemmli containing 1 mM biotin for 30 min at room temperature and then additionally boiled at 90 C for 5 min. Eluates were analyzed by immunoblotting.

### OPP incorporation assay

A total of 500000 HeLa cells per well were seeded in six-well plates a day prior to treatment. The following day, cells were incubated with either 250 nM Torin, or DMSO. Control sample was incubated with 100 µg/ml cycloheximide. For the labeling 20 µM OPP reagent (Jena Bioscience NU-931-05) was added to the media, and cells were incubated for 30 min. Following labeling, cells were washed with PBS, trypsinized and fixed with ice-cold 70% ethanol for 1 h at −20 °C. After fixation the cells were washed three times in PBS containing 0.5% Tween-20. The incorporated OPP was marked by the Alexa488 Fluor Picolyl azide using the Click-iT Plus OPP Protein synthesis assay kit (Life Technologies C10456), according to the manufacturer's instructions. In case the analysis of cells was performed via immunoblot, the click-it reaction was omitted and cell lysate was directly used for the western blot assay. The Alexa488 labeled samples were run on Guava easyCyte HT flow cytometer (Millipore) and analyzed using FlowJo software (v10). The cell population of interest was identified plotting FSC-H versus SSC-H. The single cells were gated by plotting FSC-H versus FSC-A, and the mean intensity of the Alexa488 signal within the singlets population was subjected to quantification of the OPP incorporation.

### Purification of eIF4E and associated proteins on m7GTP agarose

On the day prior to the experiment, cells were seeded in 15 cm dishes to reach 80% confluency on the following day. The next day cells were lysed with 300 µl of polysomal buffer (0,25 M HEPES pH 7.5, 50 mM MgCl2, 1 M KCl, 5% NP40, 1000 µM Cycloheximide). The lysate was clarified by centrifugation at 14000 rpm for 15 min, 4 C. The clarified lysate was incubated for 30 min with 50 µl of control agarose, (Life technologies, 26150) to remove unspecific binding to agarose. The mix was briefly centrifuged (8000 rpm, 5 min) and the supernatant was used for the subsequent assay. The sample concentration was quantified by BCA and the samples were equalized based on the observed values. A small aliquot of the sample was set aside and used as an input control. The rest of the sample was mixed with 50 µl Immobilized γ-Aminophenyl m7GTP (C-10 spacer) (Jena Bioscience, JBS-AC-155L) and incubated rotating at 4 C for 2 h. After incubation, beads were washed three times for 15 min each with 1 ml of polysome lysis buffer. Prior to the last wash, beads were transferred into a new Eppendorf tube. Elution was performed by incubation of beads with 2x Laemmli buffer at 45 C for 30 min. Analysis of co-purified proteins was performed by immunoblotting.

### Dual-luciferase translation reporter assay

All Renilla luciferase (RLuc) reporter assays are normalized to a Firefly-luciferase reporter carrying the HBB 5′UTR, except Suppl. Fig. 7F where we normalized to RLuc mRNA levels. In addition, we have a negative-control *HBB*-RLuc reporter to control for differences between RLuc and FLuc, since our test reporters are RLuc. Cells were seeded in 96-well plate format at 8000 cells per well. The cells were transfected 16–20 h after re-seeding using Lipofectamine 2000 with 100 ng of Renilla luciferase plasmid and 100 ng of firefly luciferase plasmid per well. In case reporters were co-transfection with an additional plasmid, the plasmids were transfected in the following ratio: 50 ng firefly luciferase, 25 ng Renilla, 25 ng additional plasmid. In 6 h post transfection, the treatment, where applicable, was performed by exchanging medium into one containing drug. Luciferase assays were carried out 24 h post-transfection using the Promega Dual-Luciferase assay system (Promega, E1910) following manufacturer's instructions. In case of RNA transfection, RNA constructs were transfected with Viromer mRNA (Biozyme, 230195, discontinued) or Transmessenger (Qiagen, 301525) and the dual luciferase assay was performed 6–10 h post transfection.

### Transcript distribution within polysome fractions and quantitative RT-PCR

1.5 Million HeLa cells, either WT or expressing a certain form of eIF3d, were seeded in 20 ml of growth medium in 15 cm dishes. The following day the cells were treated either with 250 nM Torin or DMSO for 2 h. The lysate was prepared as described in the ribosome profiling section above, omitting the RNaseI digestion step. The lysate was layered on a 15% to 45% sucrose gradient and centrifuged at 35000 rpm for 3 h. Fractions of 500 µl were collected. To enable normalization of transcript distribution between fractions, 10 ng/ml of exogenous in vitro transcribed Renilla luciferase mRNA was spiked into each fraction. RNA from each fraction was extracted with acid phenol as described in the ribosome profiling set-up above. The precipitated RNA pellet was resuspended in 8ul of RNase- DNase-free water and all 8 µls of RNA were used for cDNA generation with random hexamer and oligo-dT+ primers using Maxima H minus reverse transcriptase. The efficiency of Q-RT-PCR primer pairs was tested by using a serial dilution of a sample. Quantitative RT-PCR was run on QuantStudio3 with primaQUANT SYBRGreen low ROX master mix. Levels of detected RNA in each fraction were normalized to the levels of the spiked Renilla luciferase RNA in the same fraction. In case of conventional qRT-PCR, the levels of transcript were normalized to the levels of a housekeeping gene (*ACTB, RPL13A*).

For normalization of raw Renilla luciferase counts to reporters mRNA levels, a DNase 1 treatment step was introduced, to remove remaining plasmid DNA in the samples. Sequences of oligos used for Q-RT-PCR are indicated in Supplemental Data 5.

## siRNA-mediated mRNA depletion
siRNA-mediated knock-down was performed using Lipofectamine RNAiMax (Invitrogene 13778075) according to the manufacturer's instructions. Cells were seeded at density 200,000 cells per 6-well, reverse transfected with 1.5 μl of 20 μM siRNA mix and 9 μl Lipofectamine RNAiMax reagent. 72 h post transfection, cells were re-seeded in the format required for the experiment. Sequences of siRNAs are provided in Suppl. Data 6.

## Protein-RNA co-immunoprecipitation
Cells were seeded at 1,500,000 cells per 15 cm dish in 15 ml of medium one day prior to the experiment. On the day of the experiment, cells were washed once with PBS, 10 mM $MgCl_2$, 200 μM cyclohexamide and incubated with 10 ml of freshly prepared crosslinking solution (1x PBS, 10 mM $MgCl_2$, 0.025% PFA, 0.5 mM DSP, 200 μM Cycloheximide) for 15 min at room temperature with slow rocking. Following this incubation, the crosslinking solution was substituted with ice-cold quenching solution (1x PBS, 10 mM $MgCl_2$, 200 μM Cycloheximide, 300 mM Glycine), in which cells were additionally incubated for 5 min. Then cells were quickly rinsed in PBS and lysed with 150 μl of polysomal lysis buffer (0,25 M HEPES pH 7.5, 50 mM MgCl2, 1 M KCl, 5% NP40, 1000 μM Cycloheximide). The lysate was separated into two fraction - one fraction was used for the input RNA extraction, the other part was used for immunoprecipitation of the protein of interest. For EIF4E immunoprecipitation, EIF4E antibodies were pre-bound to protein A magnetic dynabeads according to the manufacturer's instructions. In the case of the Flag-eIF3d precipitation, anti-flag-beads were used (Sigma, A2220-5ml). The lysate was incubated with the beads for 2 h, rotating at 4 °C. To remove the unbound fraction, beads were washed three times with wash buffer (20 mM Tris pH 7.4, 10 mM MgCl2, 140 mM KCl, 1% NP40). At the last wash, the beads were moved into a new tube. To reverse crosslinking, the beads were incubated in 500 μl of wash buffer with 55 μl (1/9$^{th}$ of volume) of crosslink-removal solution (10% SDS, 100 mM EDTA, 50 mM DTT) and 600 μl Acid-Phenol Chloroform (Ambion). This mix was incubated at 65 °C for 45 min with shaking at 1300 rpm, followed by 5 min incubation on ice. Then the mix was spun at 20.000 g for 5 min at room temperature, and the supernatant was used for acid-phenol chloroform extraction as described above. Isolated RNA was used either for library preparation or reverse transcription followed by qRT-PCR analysis. In case of detection of reporters with various 5′UTRs bound to eIF3d, DNase treatment was introduced prior the reverse transcriotion. DNase treatment was performed using the Turbo DNA-free kit (Invitrogene, AM1907), according to manufacturer's instructions. To obtain the protein fraction from the IP, the organic phase obtained after crosslinking reversal was subjected to ethanol, isopropanol precipitation. For this 300 μl of ethanol and 1.5 ml of isopropanol were added and the resulting solution was incubated at −20 °C for 1 h. The precipitated proteins were pelleted by centrifugation at 20.000 g for 20 min. The pellet was washed twice with 95% ethanol, 0.3 M guanidine HCl, and dried, followed by resuspension in 1x Laemmli buffer.

## Dot-blot detection of m6A
The concentration of purified mRNA was estimated with the Qubit Broad Range RNA assay kit (Life Technologies, Q10211). An equal amount of RNA was dropped on positively charged nylone membrane (Life Technologies, AM10102) followed by UV crosslinking. The membrane was then incubated with anti-m6A antibodies overnight, followed by conventional western blot protocol.

## Reporting summary
Further information on research design is available in the Nature Portfolio Reporting Summary linked to this article.

## Data availability
The data supporting the findings of this study are available from the corresponding authors upon request. All sequencing data have been deposited at NCBI Geo (GSE243708). Source data are provided with this paper.

## Code availability
All custom software used in this study is available at GitHub: https://github.com/aurelioteleman/Teleman-Lab.

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

## Acknowledgements
We thank the DKFZ Genomics and Proteomics core facility for next-generation sequencing.

## Author contributions
M.R. and M.N. performed the experiments. M.R., M.N., and A.A.T. designed the experiments, analyzed data, and wrote the manuscript.

## Funding

## Competing interests
The authors declare no competing interests.
