## [Peer Review File · Nature Communications]

eIF4E-independent translation is largely eIF3d-dependentREVIEWER COMMENTS

Reviewer #1 (Remarks to the Author):

In this publication Teleman and colleagues have identified mRNAs that are translated in an eIF4E-independent manner when mTORC1 is downregulated in activity. They find that the recently discovered alternate cap-dependent mRNA translation mechanism that involves non-canonical cap binding protein eIF3d is involved. While the work is well done, the conclusions largely restate the obvious, much of which is now known. In addition, the manuscript is more like a review than a novel research article. In fact, some of the figures with experimental figures reproduce similar experiments of those already published. Publications studies from the Lee and Schneider groups have previously identified eIF3d as a novel cap binding protein, and the Schneider group has shown that eIF3d with translation factor eIF4G2/DAP5 translates mRNAs in a cap dependent manner when mTORC1 activity is downregulated. Both groups report mRNAs that are eIF3d and DAP5/eIF3d-dependent. This study is more confirmative of previous studies than novel.

Comments

1. Figure 1 and 2 combined comments. In Figure 1, "mTORC1 inhibition does not completely block translation" the authors state what has been known for many years and shown in many published studies. Data in this figure should be a supplementary figure. It is also stated that 4E-BP/eIF4A overexpression phenocopies the eIF4E inhibition observed upon strong mTORC1 inhibition, which is indeed true in the cap immunoprecipitation (IP). However, polysomes from Torin treated cells are dramatically reduced, whereas the eIF4A mutant polysomes are similar to control cells. Moreover, both treatments show an increase on the 80S monosome peak, but a reduction in polysomes is not obvious for the mutant. How can this be explained? There is something off here that needs further study.

2. Figure 2A/C. GAPDH and RPL13 are not changed in the volcano-plot of the eIF4A mutant, whereas their expression in the Torin polysome fractions are significantly changed. If it indicates a major transcriptional differential expression rather than translational, then the conclusions are not correct.

How was DYNCH1 detected? It is not present in the scatterplot.

Despite all mRNAs being resistant, the pattern of loading on polysomes are not consistent with this conclusion. This needs to be sorted out.

3. Figure 2D. The translation efficiency (TE) of eIF4E and eIF4G1 in the volcano plot is shown as unchanged. However, it is reduced significantly in AHA IPed proteins. How is that explained? These data are not consistent.

Though an analysis of the UTRs is shown in the Sup Data, there is not any explanation for the behavior of the mRNAs chosen to for further study. This genome wide analysis could be used in a main figure but needs further exposition.

4. Figure 3. In the bicistronic mRNA reporter assay, none of the resistant mRNAs apart from CDKN2B and RPP25 are identified.

In Figure 3C, CDKN1B is claimed by the authors to have an IRES, but when a stem loop is placed in 5' UTR of CDKN1B, its translation is strongly downregulated, indicating it is unlikely to contain an IRES.

5. Figure 4. In Figure 4B, why is luminescence upregulated by Torin inhibition of TOR and sequestration of eIF4E availability? HBB is known to be eIF4E dependent. This needs to be sorted out.

6. Throughout the study, different mRNAs are shown for different analyses without any consistency. Is there a reason for this?

Reviewer #2 (Remarks to the Author):

In this manuscript, Roiuk et al. perform many sets of experiments to mechanistically dissect how non-canonical translation (eIF4E-independent) is regulated. They use different methods to inactivate eIF4E, and by ribosome profiling, they identify a subset of mRNAs that are resistant to eIF4E loss, remaining efficiently translated. The authors showed that those mRNAs require the

cap-binding activity of eIF3D to be translated where eIF4E is inactive.

While the study aims to address an interesting and outstanding question about the mechanisms of translational control under specific conditions, the manuscript needs major revision before being considered for publication in Nature Communications.

Main Concerns:

1. It has been well established in the field that eIF4E inhibition reduces the translation of specific mRNAs. In the introduction, the authors discuss the role of eIF4E under different physiologically important stress conditions (hypoxia, low nutrients, etc.); however, they use non-physiological conditions to mimic such states: Overexpression of a constitutively active 4EBP mutant, treatment with Torin, and eIF4E knock-down. There is a lack of focus with the authors constantly switching between these approaches to reduce eIF4E activity for each experiment, making the manuscript very difficult to follow and read. Importantly, the results are not consistent between different conditions (Ex: Fig 2, Torin vs 4EBP-4A overexpression), and therefore very difficult to comprehend.

2. The main novelty of the study is the identification of specific transcripts that are translated by eIF3D, although this important point is not well-addressed. The authors should provide some insight about why those transcripts will be translated by eIF3D, instead of eIF4E. For example: Are those transcripts still dependent on eIF3D when eIF4E is not limited? In which physiological condition is this regulation more important? Other than the length of the 5'UTRs, do those transcripts have any common features? The analyses of the select 5'UTRs included in Supp Fig. 17 are tantalizing but do not provide any substantial new insights.

3. The manuscript contains so many experiments that are not relevant to their hypotheses, distract from the main findings and/or are not explained in the context. It is very confusing for the reader. Key examples are: (1) The graphs are almost impossible to read and interpret (Fig 6, Fig S13A). (2) The screen results provide no new insights (Supp Fig 13 & 14). (3) The series of m6A experiments are confusing to a broad audience, while not providing any additional information about eIF4E independent translation (Supp Fig 9C-F).

Other Concerns:

1. Data quality and experimental design:

a. Many of the presented polysome traces are not of sufficient quality for publication (Supp Fig. 15C, Supp Fig. 16E).

b. Why do the authors deplete eIF4E3 when it is clearly not expressed at the protein level? How can the reader reliably interpret the results in Supp Fig. 11F?

2. The authors need to clarify which experiments are conducted with the full-length constitutively active 4EBP1 construct and which are done with the TOS domain deletion. For example, is the reader to understand that in Fig. 5E the authors are comparing the ribo-seq with the full length 4EBP1 with the IP-seq data from the TOS deletion?

3. The authors must clarify how their luciferase experiments are normalized. For example, the authors show that the HBB 5'UTR is sensitive to eIF4E levels (at least in some settings, e.g. Supp Fig. 7), but then appear to be using it as a control for normalization (e.g. Fig 3).

Reviewer #3 (Remarks to the Author):

In this manuscript, authors show that in conditions of inactivation of mTORC1 and thereby inactivation of eIF4E1, some mRNAs are still efficiently translated. They found that these mRNAs preferentially release eIF4E1 when eIF4E1 is inactive and bind instead to eIF3D via its cap-binding pocket. eIF3D then enables these mRNAs to be efficiently translated due to its cap-binding activity.

The manuscript is an interesting and comprehensive piece of work. It is well-written, logically presented, includes the appropriate controls, and the conclusions that are made are supported by the experimental results shown. Thus, this work seems suitable for publication in Nature Communications. Nevertheless, there are two points that could be further investigated and corresponding data added to the manuscript:

Authors could better dissect the mechanism by which eIF3D is involved in eIF3E-independent translation initiation. Also, it would be interesting to unequivocally show what causes the selective

binding of eIF3D to some mRNAs, while some others are resistant, in conditions of eIF4E inhibition. These two points would improve the manuscript.

Reviewer #1

In this publication Teleman and colleagues have identified mRNAs that are translated in an eIF4E-independent manner when mTORC1 is downregulated in activity. They find that the recently discovered alternate cap-dependent mRNA translation mechanism that involves non-canonical cap binding protein eIF3d is involved. While the work is well done, the conclusions largely restate the obvious, much of which is now known. In addition, the manuscript is more like a review than a novel research article. In fact, some of the figures with experimental figures reproduce similar experiments of those already published. Publications studies from the Lee and Schneider groups have previously identified eIF3d as a novel cap binding protein, and the Schneider group has shown that eIF3d with translation factor eIF4G2/DAP5 translates mRNAs in a cap dependent manner when mTORC1 activity is downregulated. Both groups report mRNAs that are eIF3d and DAP5/eIF3d-dependent. This study is more confirmative of previous studies than novel.

Although our study agrees with the published literature, it takes a different angle from what is already published, by showing that the majority of translation that occurs when eIF4E1 is inhibited is eIF3d dependent. Indeed, it was known that eIF3d can bind the cap and drive translation independently of eIF4E1, but this could have been just one of many mechanisms that become activated when eIF4E1-dependent translation turns off. A large number (>30) of cap-binding proteins have been described. In addition, eIF4E1 has two paralogs, 4E2 and 4E3, and there is an entire family of Larp proteins which can also bind the cap. Furthermore, IRES-dependent translation has been described, as well as m6A-dependent translation. So a certain fraction of eIF4E1-independent translation could have been eIF3d-dependent, and the rest could have been dependent on other mechanisms. We systematically profile all the eIF4E1-independent translation, and we systematically test all the other possible mechanism, to arrive at the conclusion that mRNA translation is basically either eIF4E1-dependent or eIF3d-dependent. We believe this is an important understanding.

In addition, we have now added additional novelty to the revised manuscript:

- We now dissect the two functional elements present in the RPP25 and CDKN1B 5'UTRs to understand what characteristics are needed to render translation of an mRNA independent of eIF4E (Suppl. Fig. 18-19). Our data indicate that in the case of CDKN1B, the functional element is not a hairpin, as has been described by the Cate and Lee labs for cJun and ALKBH5. This suggests that there may be more than one mechanism how eIF3d is recruited to an mRNA. We identify a critical 10-nt C/T-rich sequence needed for eIF3d dependence. We also study where this element needs to be placed within the 5'UTR, and discover that it needs to be <300nt from the cap, whereas the distance to the ORF start codon is not critical. In sum, this work provides new insights into the sequence requirements for eIF3d-dependent translation.

- Importantly, we discover that also under non-stressed conditions, translation of these "eIF4E-resistant mRNAs" is boosted by eIF3d: when eIF3d is knocked down, the activity of luciferase reporters carrying these 5'UTRs drops more strongly than the activity of negative control reporters or the activity of the normalization control reporter.

In the non-stressed condition, however, the cap-binding capacity of eIF3d is not needed (Fig. 6A). Impressively, if we multimerize the 108-nt functional element of CDKN1B, this can boost translation of an mRNA very strongly (Fig. 6B-C) in an eIF3d-dependent manner (Fig. 6D) !

We therefore propose a model whereby recruitment of eIF3d to an mRNA boosts its translation in both stressed and non-stressed conditions. The cap-binding capacity of eIF3d only becomes needed if eIF4E is inactivated (Fig. 7). We believe these are some important new conceptual insights.

Comments

1. Figure 1 and 2 combined comments. In Figure 1, "mTORC1 inhibition does not completely block translation" the authors state what has been known for many years and shown in many published studies. Data in this figure should be a supplementary figure.

This figure is mainly setting up and characterizing the system that we use in the rest of the paper. As suggested, we have shifted it to a supplementary figure.

It is also stated that 4E-BP/eIF4A overexpression phenocopies the eIF4E inhibition observed upon strong mTORC1 inhibition, which is indeed true in the cap immunoprecipitation (IP). However, polysomes from Torin treated cells are dramatically reduced, whereas the eIF4A mutant polysome are similar to control cells.

This is due to two effects:

1. A normalization / visual effect. The original figure showing the polysome profile upon 4E-BP1-4A expression is now shifted to Suppl. Fig. 1I. In this figure, one can see that the 80S peak is much larger in the 4E-BP condition than the control condition, and the polysomes are mildly reduced. If one were to normalize the two graphs to the 80S peak, the polysome peaks would be reduced by 50%. This is why we quantified the polysome-to-monosome ratio (now Suppl. Fig. 1J), which shows a 50% drop, similar in magnitude to the drop in puromycin incorporation caused by Torin (Suppl. Fig. 1A). In response to one of the other reviewer comments, we replaced what is now main Figure 1B, which was previously a Torin treatment, with new polysome gradients from

4E-BP1-4A expression, and in this figure the effect of 4E-BP expression on the polysomes is stronger.

2. That said, TORC1 inhibition also had additional effects (on LARP1, eIF4B and RpS6 activity) which 4E-BP1-4A expression does not do, so 4E-BP1-4A expression is a more specific manipulation to inhibit eIF4E-dependent translation. We now consistently use 4E-BP expression as our main manipulation throughout the manuscript.

For this reason, we wrote

"4E-BP-4A overexpression phenocopies the eIF4E inhibition observed upon strong mTORC1 inhibition.

Moreover, both treatments show an increase on the 80S monosome peak, but a reduction in polysomes is not obvious for the mutant. How can this be explained? There is something off here that needs further study.

As discussed above, this is just an effect of how the polysome curves are normalized to each other for display in the figure - the polysome-to-monosome ratio is invariant and important here.

2. Figure 2A/C. GAPDH and RPL13 are not changed in the volcano-plot of the eIF4A mutant, whereas their expression in the Torin polysome fractions are significantly changed. If it indicates a major transcriptional differential expression rather than translational, then the conclusions are not correct.

(Note, this is now Figure 1).

This is again an issue of normalization:

- Both the riboseq (panel A) and the polysome Q-RT-PCR (panel C) normalize away transcriptional changes. In the volcano plot, the riboseq is normalized to total RNA, yielding translation efficiency. In the polysome plots, the values are shown as % of total RNA. So the difference observed by the reviewer is not due to transcription.

- However, the volcano plot also normalizes away global changes in translation, setting the average fold-change in translation efficiency to 1, whereas the polysome Q-RT-PCR does not. We know there is a global drop in translation of roughly 50% upon 4EBP expression (Suppl. Fig. 1H, J) so on average the genes that do not change in the volcano plot should shift towards monosomes in the polysome gradient, which is what we see in Fig. 1C, and what the reviewer points out.

Note that in response to another reviewer comment, we replaced Fig. 1B-C with polysome gradients from cells +/- 4E-BP1-4A expression and moved the original polysomes +/- Torin treatment to Suppl. Fig. 3B-C, but the same logic applies to both polysome experiments and the results are the same.

How was DYNCIH1 detected? It is not present in the scatterplot.

We thank the reviewer for noticing this - we forgot to label the DYNC1H1 dot in the original volcano plot. We have now fixed this. The translation efficiency of DYNC1H1 increases roughly 2-fold, hence the behavior on the polysome gradient is consistent with the volcano plot.

Despite all mRNAs being resistant, the pattern of loading on polysomes are not consistent with this conclusion. This needs to be sorted out.

We are not sure we understand this comment. Although the polysome loading varies between genes, partly due to mRNA length (the longer mRNAs can load more ribosomes) and partly due to translation rates on the mRNAs, nonetheless, the ones that are resistant in the footprinting data do not shift towards monosomes in the polysome gradients, which is what one would expect.

3. Figure 2D. The translation efficiency (TE) of eIF4E and eIF4G1 in the volcano plot is shown as unchanged. However, it is reduced significantly in AHA IPed proteins. How is that explained? These data are not consistent.

Please see the explanation above. This is because the data in the volcano plot are normalized so that the average fold-change in translation efficiency is 1 (since ribosome profiling normalizes away global changes in translation rates). However, 4E-BP-1 expression causes a drop in global translation to 50% (Suppl. Fig. 1H, J). Hence, genes such as eIF4E and eIF4G1 that have an unchanged TE on the volcano plot are actually dropping in translation by 50%, and are therefore expected to drop in the BONCAT assay show in Fig. 1D-E.

Though an analysis of the UTRs is shown in the Sup Data, there is not any explanation for the behavior of the mRNAs chosen to for further study. This genome wide analysis could be used in a main figure but needs further exposition.

We selected all the top hits from the riboseq (ie highest log₂(FC) in TE upon 4E-BP1-4A expression) for further analysis in Fig 2A. For some of them, however, we did not get a PCR product when trying to clone the 5'UTR, so a few of the top hits are missing.

Otherwise, we used all the top hits that we were able to clone to test resistance to 4E-BP1-4A. From all subsequent assays from Fig 2B onwards, we then selected the ~12 reporters that showed consistent induction as 'resistant reporters', as described in the text.

We assume that by "genome wide analysis" the reviewer is referring to Suppl. Fig 6 regarding features of the resistant mRNAs? This analysis, however, does not yield any detailed insights. Instead, we now provide significant new insights into the functional element that impart eIF3d-dependent translation, which we have now included as main Fig. 6 and Suppl. Fig. 18-19.

4. Figure 3. In the bicistronic mRNA reporter assay, none of the resistant mRNAs apart from CDKN2B and RPP25 are identified. In Figure 3C, CDKN1B is claimed by the authors to have an IRES, but when a stem loop is placed in 5' UTR of CDKN1B, its translation is strongly downregulated, indicating it is unlikely to contain an IRES.

We agree with the reviewer- that is exactly our point. We do not believe any of these 5'UTRs contain functional IRESs. (Indeed this section in the results is entitled "Resistance to eIF4E inhibition does not require IRES activity".) We have now tried to write this conclusion more clearly in the text:

"Thus, most of the translation on these "resistant" 5'UTRs appears to be IRES-independent and 5'-end dependent."

5. Figure 4. In Figure 4B, why is luminescence upregulated by Torin inhibition of TOR and sequestration of eIF4E availability? HBB is known to be eIF4E dependent. This needs to be sorted out.

Fig 3B (old Fig. 4B) is a 'zoom in' of the A-capped reporters in Fig 3A, which have very low luminescence compared to the capped reporters, as would be expected. Since in Fig 3B, the reporters are A-capped, and hence they are not recognized by eIF4E, we would not expect them to drop upon TOR inhibition, since eIF4E is not binding them. In addition, since the data are normalized to a capped FLuc reporter, which does decrease upon TOR inhibition, the ratio of (A-capped RLuc reporters) to (capped Fluc normalization control) goes up. These results are exactly what one would expect, and fully consistent with the literature.

6. Throughout the study, different mRNAs are shown for different analyses without any consistency. Is there a reason for this?

There was no particular reason for this. We have now done a substantial amount of work to include all the reporters in all the relevant figures, and have now updated Fig. 3A, Fig. 3B, Fig. 5A, Fig. 6A and Fig. 6C.

Reviewer #2

In this manuscript, Roiuk et al. perform many sets of experiments to mechanistically dissect how non-canonical translation (eIF4E-independent) is regulated. They use different methods to inactivate eIF4E, and by ribosome profiling, they identify a subset of mRNAs that are resistant to eIF4E loss, remaining efficiently translated. The authors showed that those mRNAs require the cap-binding activity of eIF3D to be translated where eIF4E is inactive.

While the study aims to address an interesting and outstanding question about the mechanisms of translational control under specific conditions, the manuscript needs major revision before being considered for publication in Nature Communications.

Main Concerns:

1. It has been well established in the field that eIF4E inhibition reduces the translation of specific mRNAs. In the introduction, the authors discuss the role of eIF4E under different physiologically important stress conditions (hypoxia, low nutrients, etc.); however, they use non-physiological conditions to mimic such states: Overexpression of a constitutively active 4EBP mutant, treatment with Torin, and eIF4E knock-down. There is a lack of focus with the authors constantly switching between these approaches to reduce eIF4E activity for each experiment, making the manuscript very difficult to follow and read. Importantly, the results are not consistent between different conditions (Ex: Fig 2, Torin vs 4EBP-4A overexpression), and therefore very difficult to comprehend.

Please note that Suppl. Fig. 3 contains data +/- glucose as a physiological stress.

We have now updated the main figures to do everything with 4EBP expression, because this is the most specific manipulation to inactivate eIF4E. The only exception are 2 experiments involving mRNA transfections - in these cases, it is technically difficult to combine mRNA transfections with 4E-BP expression: by the time 4E-BP1 is produced and active, there is little reporter mRNA left in cells and the signal we detect mostly comes from R/FLuc translated prior to 4E-BP1-4A expression, due to luciferase protein stability. We tried to sequentially transfect 4E-BP1-4A followed by transfection of mRNA reporters, but that caused strong cell toxicity. Therefore, in these two experiments we opted for mRNA transfection followed by 250nM Torin treatment.

All other data have been moved to supplement.

2. The main novelty of the study is the identification of specific transcripts that are translated by eIF3D, although this important point is not well-addressed. The authors should provide some insight about why those transcripts will be translated by eIF3D, instead of eIF4E. For example: Are those transcripts still dependent on eIF3D when eIF4E is not limited? In which physiological condition is this regulation more important? Other than the length of the 5'UTRs, do those transcripts have any common features? The analyses of the select 5'UTRs included in Supp Fig. 17 are tantalizing but do not provide any substantial new insights.

We now include a substantial amount of new data to study why certain transcripts are translated by eIF3D.

- We now dissect the two functional elements present in the RPP25 and CDKN1B 5'UTRs to understand what characteristics are needed to render translation of an mRNA independent of eIF4E (Suppl. Fig. 18-19). Our data indicate that in the case of CDKN1B, the functional element is not a hairpin, as has been described by the Cate and Lee labs for cJun and ALKBH5. This suggests that there may be more than one mechanism how eIF3d is recruited to an mRNA. We identify a critical 10-nt C/T-rich sequence needed for eIF3d dependence. We also study where this element needs to be placed within the 5'UTR, and discover that it needs to be <300nt from the cap, whereas the distance to the ORF start codon is not critical. In sum, this work provides new insights into the sequence requirements for eIF3d-dependent translation.

- Importantly, we discover that also under non-stressed conditions, translation of these "eIF4E-resistant mRNAs" is boosted by eIF3d: when eIF3d is knocked down, the activity of luciferase reporters carrying these 5'UTRs drops more strongly than the activity of negative control reporters or the activity of the normalization control reporter.

In the non-stressed condition, however, the cap-binding capacity of eIF3d is not needed (Fig. 6A). Impressively, if we multimerize the 108-nt functional element of CDKN1B, this can boost translation of an mRNA very strongly (Fig. 6B-C) in an eIF3d-dependent manner (Fig. 6D) !

We therefore propose a model whereby recruitment of eIF3d to an mRNA boosts its translation in both stressed and non-stressed conditions. The cap-binding capacity of eIF3d only becomes needed if eIF4E is inactivated (Fig. 7). We believe these are some important new conceptual insights.

Suppl. Fig. 3 shows that this regulation is relevant in a low glucose condition.

3. The manuscript contains so many experiments that are not relevant to their hypotheses, distract from the main findings and/or are not explained in the context. It is very confusing for the reader. Key examples are: (1) The graphs are almost impossible to read and interpret (Fig 6, Fig S13A). (2) The screen results provide no new insights (Supp Fig 13 & 14). (3) The series of m6A experiments are confusing to a broad audience, while not providing any additional information about eIF4E independent translation (Supp Fig 9C-F).

We believe one important finding of this manuscript is that most of the eIF4E-independent translation is eIF3d-dependent. To come to this conclusion, we excluded a whole array of other possible mechanisms including IRES-dependent translation, m6A-dependent translation, and translation via a slew of other possible cap-binding proteins. We think it is therefore important to keep this biologically-negative data.

Nonetheless, we have tried to make the reading easier by:

- removing old Suppl. Fig. 13E (a big bar graph) and replacing it with a heat map, and instead putting the data into a supplemental table in case anyone is interested.
- adjusting the coloring on the large bar graphs to make it somewhat easier to see
- keeping in Fig. 5A only the eIF3D knockdown data and shifting the rest of the knockdowns to a supplemental figure.
- adjusting the text to make it clear at the beginning of a few sections that the subsequent results will be negative (in case a reader wants to skip the rest of the section).

We hope these changes make the manuscript less confusing.

Other Concerns:

1. Data quality and experimental design:

a. Many of the presented polysome traces are not of sufficient quality for publication (Supp Fig. 15C, Supp Fig. 16E).

We have repeated and replaced the polysome gradients mentioned by the reviewer (now Suppl. Fig. 15A and Suppl. Fig. 16C).

b. Why do the authors deplete eIF4E3 when it is clearly not expressed at the protein level? How can the reader reliably interpret the results in Supp Fig. 11F?

We did this to be absolutely certain. It is possible that there are indeed low levels of eIF4E3 protein, but we could not detect them because the antibody is not of sufficient sensitivity. The mRNA is there (at low levels) when we do Q-RT-PCR, and these levels

are reduced by the knockdown, so it's a specific signal. Hence these experiments allowed us to confidently rule out that eIF3E3 is playing a role here.

2. The authors need to clarify which experiments are conducted with the full-length constitutively active 4EBP1 construct and which are done with the TOS domain deletion. For example, is the reader to understand that in Fig. 5E the authors are comparing the ribo-seq with the full length 4EBP1 with the IP-seq data from the TOS deletion?

All data presented up to main Figure 1 are obtained with full-length 4E-BP1-4A, and all others are with the deltaTOS version, except Figure 4 in order to directly compare the pull-down data in Fig. 4 to the riboseq in Fig 1.

After presenting Figure 1 in the Results section, we added the sentence

"Thus, we used 4E-BP-4A Δ TOS for all further experiments in this manuscript, unless otherwise stated."

and in the legend of Figure 4 we wrote that this was done with full-length 4E-BP1-4A.

3. The authors must clarify how their luciferase experiments are normalized. For example, the authors show that the HBB 5'UTR is sensitive to eIF4E levels (at least in some settings, e.g. Supp Fig. 7), but then appear to be using it as a control for normalization (e.g. Fig 3).

All Renilla luciferase (RLuc) assays are normalized to an HBB-Firefly-luciferase reporter, except Suppl. Fig. 7E (mentioned by the reviewer) where we normalized to mRNA levels. In addition, we have a negative-control HBB-RLuc reporter to control for differences between RLuc and FLuc (since our test reporters are RLuc). In this way the experiment is fully controlled. This means that HBB-RLuc normalized to HBB-FLuc should not change in response to manipulations, whereas HBB-RLuc normalized to mRNA levels can indeed change due to global changes in translation levels.

We have added this information to the methods.

Reviewer #3

In this manuscript, authors show that in conditions of inactivation of mTORC1 and thereby inactivation of eIF4E1, some mRNAs are still efficiently translated. They found that these mRNAs preferentially release eIF4E1 when eIF4E1 is inactive and bind instead to eIF3D via its cap-binding pocket. eIF3D then

enables these mRNAs to be efficiently translated due to its cap-binding activity.

The manuscript is an interesting and comprehensive piece of work. It is well-written, logically presented, includes the appropriate controls, and the conclusions that are made are supported by the experimental results shown. Thus, this work seems suitable for publication in Nature Communications.

We thank the reviewer for the supportive words.

Nevertheless, there are two points that could be further investigated and corresponding data added to the manuscript:

Authors could better dissect the mechanism by which eIF3D is involved in eIF3E-independent translation initiation.

We thank the reviewer for prodding us into doing these experiments, which we have added to the manuscript and have yielded some interesting new insights:

- We now dissect the two functional elements present in the RPP25 and CDKN1B 5'UTRs to understand what characteristics are needed to render translation of an mRNA independent of eIF4E (Suppl. Fig. 18-19). Our data indicate that in the case of CDKN1B, the functional element is not a hairpin, as has been described by the Cate and Lee labs for cJun and ALKBH5. This suggests that there may be more than one mechanism how eIF3d is recruited to an mRNA. We identify a critical 10-nt C/T-rich sequence needed for eIF3d dependence. We also study where this element needs to be placed within the 5'UTR, and discover that it needs to be <300nt from the cap, whereas the distance to the ORF start codon is not critical. In sum, this work provides new insights into the sequence requirements for eIF3d-dependent translation.

- Importantly, we discover that also under non-stressed conditions, translation of these "eIF4E-resistant mRNAs" is boosted by eIF3d: when eIF3d is knocked down, the activity of luciferase reporters carrying these 5'UTRs drops more strongly than the activity of negative control reporters or the activity of the normalization control reporter.

In the non-stressed condition, however, the cap-binding capacity of eIF3d is not needed (Fig. 6A). Impressively, if we multimerize the 108-nt functional element of CDKN1B, this can boost translation of an mRNA very strongly (Fig. 6B-C) in an eIF3d-dependent manner (Fig. 6D) !

We therefore propose a model whereby recruitment of eIF3d to an mRNA boosts its translation in both stressed and non-stressed conditions. The cap-binding capacity of eIF3d only becomes needed if eIF4E is inactivated (Fig. 7). We believe these are some important new conceptual insights.

Also, it would be interesting to unequivocally show what causes the selective binding of eIF3D to some mRNAs, while some others are resistant, in conditions of eIF4E inhibition. These two points would improve the manuscript.

We now provide in Suppl. Fig. 17F data showing that the functional elements in the RPP25 5'UTR and in the CDKN1B 5'UTR which mediate resistance to eIF4E-inactivation also mediate binding to eIF3d. In the case of CDKN1B, we identify a 10nt sequence (TGGTCCCCTC) which is critical for this resistance. We do not find this sequence in the RPP25 functional element, which instead is G/C rich, resembling an eIF4G2 binding site, suggesting that there may be different modes of recruiting eIF3d to a 5'UTR.

REVIEWERS' COMMENTS

Reviewer #1 (Remarks to the Author):

The authors have responded to reviews in a serious way, by conducting additional extensive studies to clarify and add novel content to their study. They have addressed all of the previous issues raised by this reviewer.

Reviewer #2 (Remarks to the Author):

The authors have sufficiently addressed all of the previously raised concerns, and the revised manuscript is acceptable for publication in Nature Communications.

The authors are commended for improving the clarity, graphical presentations, and overall novelty of their work. In particular, the simplification and focus on their 4EBP1 repression system have greatly enhanced the interpretability of the manuscript for a more general audience. Additionally, the authors' efforts to clarify the graphical representation of the data are appreciated.

Finally, the substantial new data exploring the possible regulatory elements within the 5'UTRs of the "eIF4E resistant" genes enhance the novelty and impact of the work. The identification of the 10-nt C/T-rich eIF3d-dependent functional element presents new insights that will be fruitful for future research.

There are only two minor concerns that need to be addressed:

1. Please label Supp Fig15A with +eIF3D(WT) instead of eIF3.
2. In Fig 5E, the authors use ACTB 5'UTR instead of HBB, which has been used throughout the manuscript, and importantly, ACTB 5'UTR is as sensitive as all resistant-5UTRs to eIF3d mutant PD. It is also confusing why the authors do not see a significant increase in eIF3D binding upon 4EBP-4A overexpression. Also, it seems that the activity of the eIF3D mutant is globally less efficient, as all the reporters show significantly less binding. The authors should clarify this and redo the analysis or remove the data.

Reviewer #3 (Remarks to the Author):

The new version of the manuscript has been improved, and authors have appropriately replied to my concerns. Thus the manuscript can be accepted for publication in Nature Communications.

Reviewer #1 (Remarks to the Author):

The authors have responded to reviews in a serious way, by conducting additional extensive studies to clarify and add novel content to their study. They have addressed all of the previous issues raised by this reviewer.

Thank you !

Reviewer #2 (Remarks to the Author):

The authors have sufficiently addressed all of the previously raised concerns, and the revised manuscript is acceptable for publication in Nature Communications.

The authors are commended for improving the clarity, graphical presentations, and overall novelty of their work. In particular, the simplification and focus on their 4EBP1 repression system have greatly enhanced the interpretability of the manuscript for a more general audience. Additionally, the authors' efforts to clarify the graphical representation of the data are appreciated.

Finally, the substantial new data exploring the possible regulatory elements within the 5'UTRs of the "eIF4E resistant" genes enhance the novelty and impact of the work. The identification of the 10-nt C/T-rich eIF3d-dependent functional element presents new insights that will be fruitful for future research.

Thanks !

There are only two minor concerns that need to be addressed:

1. Please label Supp Fig15A with +eIF3D(WT) instead of eIF3.

Done.

2. In Fig 5E, the authors use ACTB 5'UTR instead of HBB, which has been used throughout the manuscript, and importantly, ACTB 5'UTR is as sensitive as all resistant-5UTRs to eIF3d mutant PD. It is also confusing why the authors do not see a significant increase in eIF3D binding upon 4EBP-4A overexpression. Also, it seems that the activity of the eIF3D mutant is globally less efficient, as all the reporters show significantly less binding. The authors should clarify this and redo the analysis or remove the data.

Please note that this is an IP of endogenous mRNAs, this is not a reporter assay. Since HeLa cells do not express HBB, we cannot assay its binding to eIF3d. Instead, we use 3 different negative control genes – GAPDH, RPL13A and ACTB. They all behave the same way, so the result is not specific to ACTB. The reduced binding of all mRNAs (both the resistant ones and the three negative controls) to a cap-binding deficient eIF3d upon eIF4E inactivation is exactly what one would expect – this is reflecting the canonical function of eIF3d as part of the eIF3 complex that is impaired when eIF4E function is inhibited and hence canonical initiation is reduced.

Reviewer #3 (Remarks to the Author):

The new version of the manuscript has been improved, and authors have appropriately replied to my concerns. Thus the manuscript can be accepted for publication in Nature Communications.

Thanks !